# WHERE DOES IN-CONTEXT MACHINE TRANSLATION HAPPEN IN LARGE LANGUAGE MODELS?

## ABSTRACT

Self-supervised large language models have demonstrated the ability to perform Machine Translation (MT) via in-context learning, but little is known about where the model performs MT with respect to prompt instructions and demonstration examples. In this work, we attempt to characterize the region in layer-wise attention heads where GPT models transition from in-context learners to translation models. Through a series of layer-wise context-masking experiments on GPTNeo2.7B and Bloom3B, we demonstrate evidence of a "task recognition" point where the translation task is encoded into the input representations and attention to context is no longer necessary. Our layer-wise fine-tuning experiments indicate that the most effective layers for MT fine-tuning are the layers critical to task recognition. Next, we examine redundancy in layers following task recognition, observing that masking these later layers does not hurt performance significantly. Finally, we train discrete attention head gates with $L_0$ regularisation and find evidence that the most pruneable heads occur after task recognition.

## 1 INTRODUCTION

*In-context learning* (ICL) refers to the phenomenon in which large generative pretrained transformers (GPTs) perform novel tasks with no gradient updates when shown task examples or descriptions in their context (Brown et al., 2020; Bommasani et al., 2021). While in-context learning in GPT models appears to be generally applicable to any natural language task, machine translation (MT) has recently received increasing attention in this paradigm due to its potential to rapidly adapt to different languages and domains. Although in-context MT has yet to reach parity with supervised neural MT models, it's off-the-shelf translation performance is comparatively strong and suggests a promising direction for the future of MT (Hendy et al., 2023; Garcia et al., 2023). Prior work on in-context MT has focused on *prompt-engineering*, treating GPT models as black boxes by focusing on which examples to provide in-context. Agrawal et al. (2022) apply similarity-based retrieval to select in-context examples, while Sia & Duh (2023) suggest a coherence-based approach. However, these works apply surface level interventions leaving the internal mechanism of MT in GPT models largely not understood.

In this work, we ask **where does in-context Machine Translation occur** in GPT models? We conduct an initial exploration into locating self-attention layers responsible for in-context MT in two open-source GPT models. Using causal masking over different parts of the context we demonstrate that there exists a "task-recognition" point after which attention to the context is no longer necessary. Next, we observe that very lightweight fine-tuning of LoRA parameters (Hu et al., 2021) are most effective at the earlier layers of the model, layers we conjecture are responsible for task location, corroborating prior work on the importance of prompts (Hendy et al., 2023).

Having identified the layers in which "task recognition" occurs, we study the extent to which subsequent layers are *redundant*. Simple layer-wise masking shows that removing attention around the "task-recognition" layers can cause the model to fail to perform translation all-together, whereas layers towards the end of the model are much more redundant. We further investigate the extent of MT *task redundancy* using differentiable $L_0$ regularisation to train discrete attention head gates (Section 6.2). We find that around 10% of the attention heads can be masked, which fundamentally differs from the literature in supervised NMT where attention heads are highly specialised for MT (Voita et al., 2019; Michel et al., 2019; Behnke & Heafield, 2021).

## 2 BACKGROUND

**In-Context Learning** was first demonstrated by Brown et al. (2020) who showed that GPT-3 could be used to perform a huge variety of tasks without any task-specific parameters or training, by conditioning the model's generation on a *prompt* which included a few labeled examples of the task of interest. Since then, interest in using GPT models for ICL has grown significantly, with several recent works introducing methods such as instruction-tuning (Sanh et al., 2022; Wang et al., 2022) or chain-of-thought prompting (Wei et al., 2022) to improve downstream ICL accuracy.

Ostensibly, ICL can work for nearly any task that can be defined or described in natural language, and therefore has potential for incredibly broad impact. However, ICL can often still underperform supervised fine-tuning (Bhatia et al., 2023), prompting research in analyzing the mechanisms underlying ICL. One line of work studies in-context learning with *linear* functions, typically linear regression, characterizing the learnability of these functions with ICL (Li et al., 2023; Garg et al., 2022) and even the learning algorithm a transformer uses Akyürek et al. (2022); Dai et al. (2023); von Oswald et al. (2023). A second body of work suggests that in-context learning locates *existing* latent concepts (tasks) which have been *already learnt* during pretraining (Xie et al., 2021; Wies et al., 2023). Finally, Wei et al. (2023) suggest that model size may change the mechanisms behind ICL from latent inference to actual learning algorithms as size increases. Our work which focuses on Machine Translation, fits into this recent chain of work by demonstrating that there exists a point in the model's *layers* where the task has been located.

**In-Context Machine Translation** While GPT models are strong few-shot learners, their pre-training data is historically dominated by English, limiting their ability to perform translation tasks (Hendy et al., 2023). Lin et al. (2022) find that an explicitly multilingual GPT significantly outperforms traditional english models such as GPT-3, and Garcia et al. (2023) find that such models can even be competitive with supervised MT models in some settings. However, even with explicit multilingual pre-training, in-context MT has been found to be very sensitive to the examples used Liu et al. (2022) and their orders Lu et al. (2022a). In response, recent work focuses on how to select prompts that elicit the best downstream MT performance (Agrawal et al., 2022; Sia & Duh, 2023). However, further improvement to translation with GPT models is limited by our understanding of how MT emerges in GPT models. Our work directly analyses when, in layer representations, a GPT model becomes a translation model via in-context learning, and how that may inform decisions around parameter tuning and redundancy.

## 3 DATA AND SETTINGS

**Models** We use GPTNEO2.7B (Black et al., 2021) (32 layers × 20 heads) and BLOOM3B (Scao et al., 2022) (30 layers × 32 heads) in all of our experiments. The checkpoints we use are from the transformers library (Wolf et al., 2019). GPTNEO was trained on The PILE (Gao et al., 2020), an 825GB text dataset which consists of roughly 98% English data. Despite being mostly monolingual, The PILE contains Europarl which GPTNEO was trained on at a document level (rather than a sentence level). Conversely, BLOOM was trained on the ROOTS corpus (Laurençon et al., 2022), a composite collection of 498 datasets that were explicitly selected to be multilingual, representing 46 natural languages and 13 programming languages. To our knowledge, there has not been any reports of sentence level parallel corpora in the training datasets of these models.

**Data** We test our models using FLORES (Goyal et al., 2021) en ↔ fr which we report in the main paper, and en ↔ es in the Appendix, drawing prompt examples from the development set. To account for example selection and ordering effects,[1] all inference runs were repeated with 5 randomly sampled prompt example sets. We evaluate the generations using BLEU scores, following the implementation from Post (2018). Our prompts may consist of instructions, examples, both, or none. When no natural language instructions are used we supply the model with language indicators only, e.g. the model input will be `{L1}:{source_sentence} {L2}:` where `L1 = English` and `L2 = French` if the source and target languages are English and French respectively. Instructions are given in natural language and take the form: `Translate from {L1} to {L2}:{L1}:{source_sentence}{L2}:`. See Table 3 for an example.

---

[1] In-context learning models have been found to be sensitive to these order effects (Lu et al., 2022b).

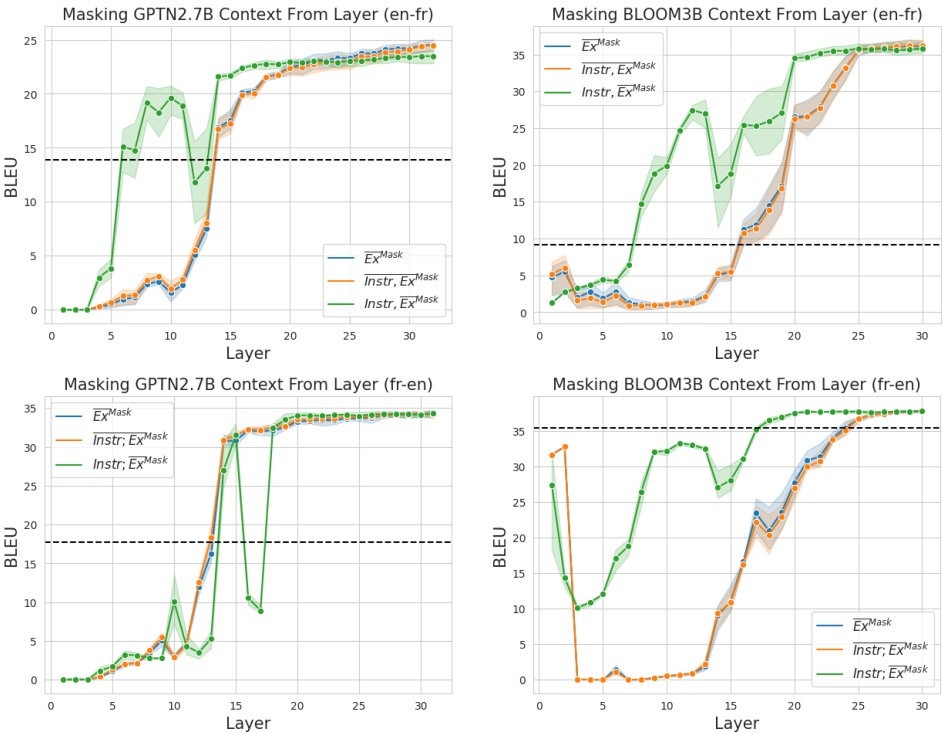

Figure 1: *Layer-from context-masking experiments* for GPTNeo and BLOOM on en ↔ fr. The graphs show translation performance when masking the in-context exemplars from the $j^{\text{th}}$ layer onwards. Different lines indicate different treatments of the instruction, as described in Figure 6. The dashed black line refers to the baseline which sees no instructions and no in-context exemplars.

| Name | Instr | Ex | |
|---|---|---|---|
| $\overline{\text{Ex}}^{Mask}$ | N | Y | En: ⋯ Fr: ⋯ En: ⋯ Fr: |
| $\text{Instr},\overline{\text{Ex}}^{Mask}$ | Y | Y | Translate En to Fr: En: ⋯ Fr:⋯ En: ⋯ Fr: |
| $\overline{\text{Instr},\text{Ex}}^{Mask}$ | Y | Y | Translate En to Fr: En: ⋯ Fr:⋯ En: ⋯ Fr: |

Table 1: For the prompt contexts of Instructions (Instr) and Examples (Ex). The overline corresponds to the yellow highlights. This refers to attention masking from future time steps to previous time steps in the causal language models. $N$ and $Y$ refer to absence and presence of either Instruction of Examples. Instr: Instructions and Ex: Examples. See also Figure 6.

## 4 WHERE DOES IN-CONTEXT MT HAPPEN?

In-context learning differs from task-specific supervised learning in that, during test time, the desired task must be identified from the context first, then executed. At what stage in the feed-forward computation does a GPT-style model transition from an in-context learner to a translation model? To explore this question, we propose *layer-from context-masking*, a causal masking method which masks out all attention weights to the context (instructions or prompts) *from* a certain layer. All masks operate from the $j$-th layer ($\ell_j$) *onwards*, i.e. masking from $\ell_{20}$ means causally masking out attention to all context positions from $\ell_{20:n_\ell}$, where $n_\ell$ is the total number of layers.

Under this causal masking treatment, after a certain layer the model must rely on the representations of the target input sentence *only* to complete the task; if the target sentence representations do not already encode the target task (translation into a specific language) then the model will fail to generate translations. In other words, the goal is to find the layers at which attention over the context at different layers may help the model "locate" the task of translation. In all experiments we mask

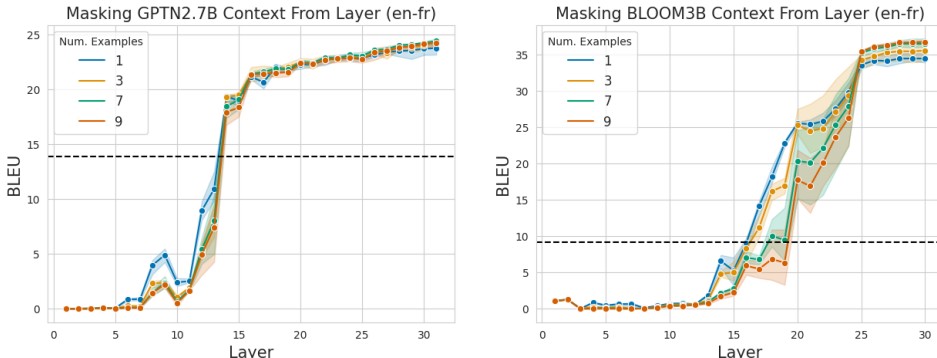

Figure 2: *Layer-from context-masking experiments* for GPTNeo and BLOOM on en $\rightarrow$ fr investigating number of prompts in the $\overline{\mathrm{Ex}}^{Mask}$ mask setting (described in Figure 6). The dashed black line refers to no instructions and no examples.

the examples provided in the context, but to control for the effect of semantic instructions, we ablate over different treatments of the instructions by removing instructions entirely ($\overline{\mathrm{Ex}}^{Mask}$), leaving them unmasked ($\mathtt{Instr}\overline{\mathrm{Ex}}^{Mask}$), or masking them together with the examples ($\overline{\mathtt{InstrEx}}^{Mask}$). See Table 1 and Figure 6 for a visual description.

The results of our experiment for en $\rightarrow$ fr and fr $\rightarrow$ en are shown in Figure 1, and additional experiments on en $\rightarrow$ es and es $\rightarrow$ en are shown in Section A.3. In both models we see a point in the model layers in which performance plateaus as a function of when the context is masked: in GPTNeo this point occurs around layer 20, and in BLOOM this point occurs around layer 25.[2] At this point, the models benefits only marginally, if at all, from attending to the context, suggesting most of the task "location" has already occurred. Prior to this point, around the middle layers of the models, moving the context mask up a layer results in a significant increase to MT performance; we conjecture that the model is locating the correct task during processing in these middle layers, after which the context is no longer necessary to perform the translation task.

For GPTNeo and BLOOM, context masking before a certain level of task-recognition results in a translation model that is often worse than the baseline which sees *no instructions or examples*. [3] We find that a series of forward computations are required to process the input, and interrupting this via our *layer from context-masking* experiments results in intermediate representations which hinder performance significantly. The picture differs for the instruction-tuned Llama7b models Touvron et al. (2023). Masking of the Examples results in little distortion, while masking of instructions results in very poor intermediate representations Figure 8.

Finally, we see that the behavior of the model is similar when no instructions are present ($\overline{\mathrm{Ex}}^{Mask}$) and when instructions are masked ($\overline{\mathtt{Instr,Ex}}^{Mask}$). However, if the model is given complete access to instructions ($\mathtt{Instr}\overline{\mathrm{Ex}}^{Mask}$), it can use the intermediate processing of examples to achieve "task recognition" earlier.

### 4.1 DOES THE NUMBER OF PROMPTS AFFECT TASK RECOGNITION?

In Section 4 we study context-masking with a fixed number of prompts. However, it is not clear if the number of prompts affects how fast, layer-wise, the model is able to recognize the task. We plot these results for en $\rightarrow$ fr in Figure 2, for both GPTNeo and BLOOM. In general, we find that the number of prompt examples has little effect on which layer the task is recognized at. While there is some variation in MT performance when the context is masked around the middle layers of the model, the final performance plateau occurs at the same layer regardless of the number of prompts.

---

[2] We note that BLOOM appears primed to perform translation into English, as it's baseline fr $\rightarrow$ en performance (with no instructions or prompts) achieves high translation accuracy already.

[3] The baseline results are not 0 because there is some minimal signal to the model based on the separator tokens e.g., 'English:' and 'French:'.

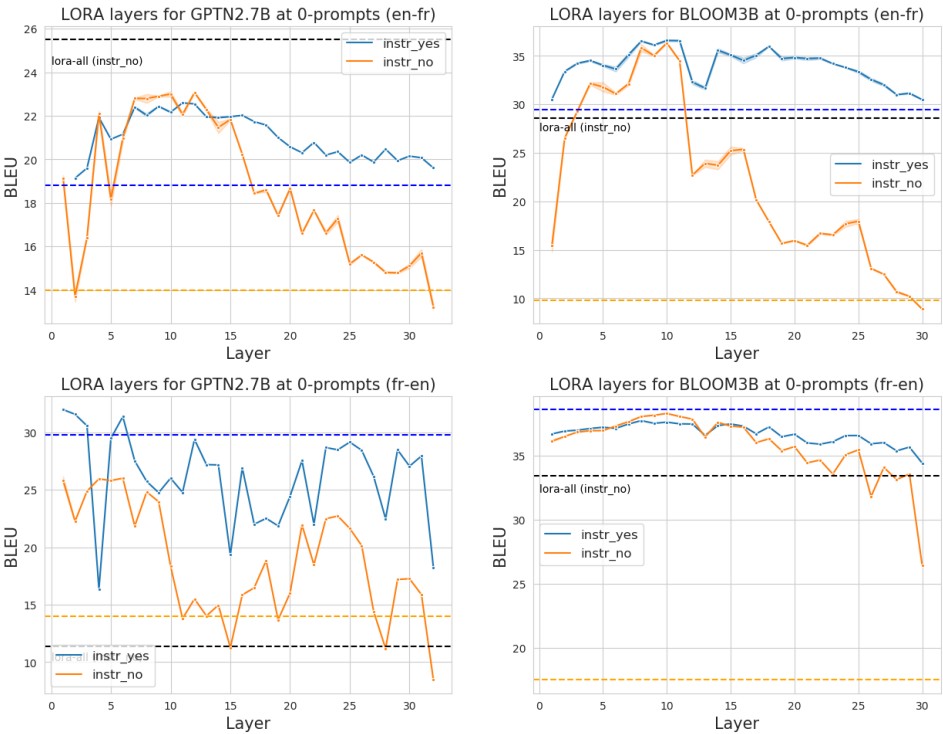

Figure 3: Performance of no-instructions trained Lora layers for GPTNeo and BLOOM on en↔fr. The dashed black line refers to training of all layers together, while the orange (test without instructions) and blue (test with instructions) dashed lines refers to no training. The layers which are most amenable to lightweight fine-tuning occur in the earlier layers before the "task recognition" point.

Overall, our context-masking experiments suggest a 3-phase process to in-context learning for MT: in the first phase [0-7 for GPTNEO, 0-12 for BLOOM], moving the mask up makes little difference in MT performance, which is close to 0, suggesting that the context has not influenced task location at all; in the second phase [8-19 for GPTNEO, 13-24 for BLOOM], shifting the mask upwards makes a large difference in MT performance, suggesting that the model has started to locate the task but can improve significantly with more processing of the context; finally, in the third phase [20-32 for GPTNEO, 25-30 for BLOOM], shifting the mask upwards again has little-to-no effect on MT performance, suggesting that the model has fully recognized the task as translation and is translation over the inputs, no longer requiring the context to interpret the task.

## 4.2 INFERENCE EFFICIENCY

We highlight the potential of speeding up inference time as a direct consequence of identifying where task recognition occurs in the model. Our results indicate that we can achieve significant speedups in inference by removing the processing of context-tokens all-together after a certain point in the model, with little impact on translation performance. Let $\ell_r$ be the $r^{\text{th}}$ layer where we can mask out the attention of the context across subsequent layers without a noticeable fall in performance. Let $k$ be the number of prompt examples, where each example consists of a pair of parallel sentences. Then, for a model with $n_\ell$ layers, the amount of processing in terms of speed and memory saved is approximately $(n_\ell - r)/n_\ell \times (k/k + 1)$. Using the example of GPTNeo (32 layers), we see from Figure 1 that the model is very close to it's ceiling score after processing the examples at layer 20 ($\ell = 20$). If we no longer need to process examples after $\ell = 20$, under a prompt size of 5 and 10, the savings are approximately 17.5% and 35% respectively.

### 4.3 THE ADAPTABILITY OF TASK LAYERS

Intuitively, the layers prior to "task recognition" should contain information about locating the MT task. To test this intuition, we further explore the adaptability of these layers by lightweight fine-tuning experiments. We trained a single Low-rank Adaptation matrix (LoRA; Hu et al. (2021)) for each layer of the output projection while keeping the rest of the network frozen.[4] The model was shown parallel sentences as input, and layers were trained with no explicit translation instructions. We split the dev set of FLORES into 800 training examples and 200 dev examples. Note that this setup is designed to tune the layers for task location. It is highly unlikely that the model can learn translation knowledge from this small amount of supervision. The LoRA layers were trained for up to 10 epochs, with $\alpha = 32, r = 32$ and dropout$= 0.1$. The cross-entropy loss was computed only on the target sentence (see Section A.7 for details) and we used the best checkpoint on the 200 held out dev examples for evaluation.

We show the results of this experiment in Figure 3; while each layer can be trained to perform better than no fine-tuning at all, tuning different layers have different impacts on performance. In particular, we find that high performing layers occur at the earlier to middle parts of the network, with the peak often occurring near the start of the "task-locating" layers from Section 4. In contrast to common fine-tuning wisdom, additional tuning on the later layers has a much smaller impact on final performance for en $\rightarrow$ fr. One likely explanation why this effect is less prominent for fr $\rightarrow$ en is that the model has less difficult locating that task.

## 5 CHARACTERISING REDUNDANCY IN IN-CONTEXT MT

Recently, Sajjad et al. (2023) found that many layers in pre-trained transformers can be dropped with little harm to downstream tasks; moreover, it is well known neural MT transformer models are known have several redundant heads which are not necessary during test time (Voita et al., 2019; Michel et al., 2019; Behnke & Heafield, 2021). However, it is not clear if the same trends hold for *in-context MT* models, and how that redundancy is related to task location versus task execution.

### 5.1 HOW CRITICAL ARE INDIVIDUAL LAYERS ACROSS THE NETWORK?

We study the contributions of individual attention-layers by performing a simple *layer-wise* masking of all self-attention heads for a single layer. When we mask layer $j$, we are masking the *attention mechanism* of layer $j$, that is the MLP of layer $j$ acts directly on the output of layer $j - 1$, rather than the output of the attention-head of layer $j$. Doing so allows us to study how *critical* each layer is, individually, to in-context Machine Translation.

We plot results for each layer of both GPTNEO and BLOOM, using all four combinations of {0 or 5} examples and {with or without} instructions in Figure 4. We identify *critical layers* as those that have a large negative impact when masked. We observe that the region of critical layers corresponds to the task locating layers in Section 4. Specifically, for GPTNEO, the largest contiguous set of layers that significanty decrease MT performance are layers 10-15; for BLOOM, those layers are 13-24. In both cases, these results suggest that the processing done in these middle layers are critically important to in-context MT.

In Section 4, we observed a very steep incline of MT performance in GPTNEO when moving context-masking from layers 10 to 15; much steeper than the layer-by-layer increase in BLOOM at any point. This suggests that the task-locating layers in GPTNEO are fewer, and therefore more heavily relied upon, whereas BLOOM may have less individually important task-location layers. This theory is corroborated by the results in Figure 4, which show that BLOOM is relatively robust to layer masking, even in the middle layers, when compared to GPTNEO, which can reach near catastrophic levels of MT performance.

Finally, we also note that GPTNEO has several layers which are critical, but not in our identified set of task-locating layers; specifically, layers 1, 2, 4, and 32 all have a significantly negative impact on

---

[4]We also experimented with the training separate Key, Query and Value LoRA Layers but found this to be less effective.

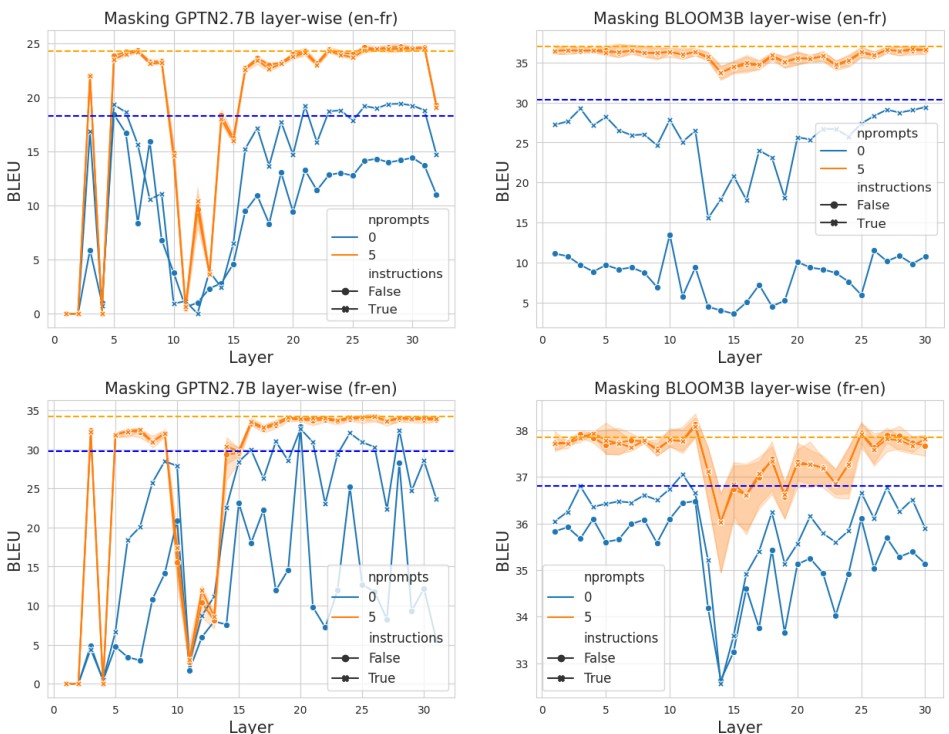

Figure 4: *Layer-wise masking* of self-attention heads for GPTNEO2.7B and BLOOM3B. The orange and blue dotted lines refer to the baselines (without masking) of 0 and 5 prompts with instructions. We observe critical layers near the middle and redundant layers towards the end of the model.

MT performance in both language directions.[5] Conversely, performance for BLOOM layer masking is robust to nearly unaffected for all layers except those in the 13-24 range.

With regard to redundancy, we find that several layers can be removed without a noticeable loss in performance. We observe that the model achieves close to baseline performance by layer-wise masking from $\ell_{15}$ for GPTNEO and for $\ell_{26}$ for BLOOM. This suggests that these later layers contain redundancy for translation. This interpretation is supported by inspecting generated output which does not change much when masking across individual layers (see Table 7 in the Appendix).

## 5.2 QUALITATIVE ANALYSIS OF LAYER-WISE MASKING

It is worth distinguishing between the model failing to translate (failure to locate the task) and degenerate outputs. When masking critical task recognition layers, there is an increased likelihood that either no output is produced, or some form of copying occurs. An example is GPTNEO, $\ell_{11}$ which copies the test source sentence for the 0-shot test setting, or the most recent sentence from the prompts for the 5-shot setting. We present examples of this using Test ID 575 and ID 600 in Table 2. Whereas for layers 1 and 4, we observe degenerate outputs such as *"B marriages {"* (Generated outputs for all layers for both GPTNeo and BLOOM are available at Section A.11).

**Interpretation of Redundant Layers** Clark et al. (2019) identify a phenomena where attention heads attend almost exclusively to delimiter and separator tokens such as [SEP], periods and commas. This is thought to act as a "no-op" as the value of such tokens in changing the current hidden representation is very small. Note that it is then possible to mask entire Transformer layers and still achieve a sensible output due to residual connections in the Transformer architecture at every layer.

---

[5]Given the extremely early location of these layers, they are likely related to the critical transformations of the input representation and we provide examples of these outputs in Section A.11.

| ID | Generated Translation | Setting | BLEU |
|----|----------------------|---------|------|
| 575 | But there are a lot of things about birds that still look like a dinosaur | 0-prompts w Instr | 2.3 |
| 600 | Hershey and Chase used phages, or viruses, to implant their own DNA into a bacterium | 0-prompts w Instr | 4.8 |
| 575 | I think it's a good idea to have a little bit of a bird in your pocket. | 0-prompts w/o Instr | 2.6 |
| 600 | The French have a saying: "The French have a saying: "The French have a saying:. | 0-prompts w/o Instr | 0.0 |
| 575 | Fondamentalement, vous afficherez des annonces pour proposer votre aide, ... | 5-prompts w Instr | 1.3 |
| 600 | Fondamentalement, vous afficherez des annonces pour proposer votre aide, ... | 5-prompts w Instr | 1.4 |
| 575 | Fondamentalement, vous afficherez des annonces pour proposer votre aide, ... | 5-prompts w/o Instr | 1.3 |
| 600 | Fondamentalement, vous afficherez des annonces pour proposer votre aide, ... | 5-prompts w/o Instr | 1.4 |

| ID | Ground Truth Translation |
|----|--------------------------|
| 575 | Mais il y a beaucoup de choses sur les oiseaux qui ressemblent encore à un dinosaure. |
| 600 | Hershey et Chase ont utilisé des phages, ou des virus, pour implanter leur propre ADN.. |

Table 2: Generated text when applying *layer-wise masking* for GPTNeo (layer 11) on en→fr. We show generated text for 0 or 5 prompts, and with or without instructions. The test IDs that were selected to be displayed were based on highest variance of BLEU score across layers for the 0-prompt with instruction setting.

### 5.3 Are layers completely redundant after task recognition?

In Section 5.1 we find a number of layers, particularly towards the end of the model, which appear to be individually redundant to the task of MT; these results might suggest that after the task-recognition point, additional processing is not necessary for in-context MT and the top layers can be removed. We conduct *layer-from masking* of all positions, in contrast to the previous experiments which masked out attention to the prompt context (instructions and examples). The results are available in Appendix: Figure 9. We find that translation improves with the inclusion of more layers cumulatively, indicating that the final layers collectively transform the latent space into the desired output, i.e. task recognition is not the same successful translation.

## 6 Are There Specialised Attention Heads for In-context MT?

In Section 4, we found that the earlier part of the model is critical for *task location* from the prompt context, and in Section 5.1 we found both critical and redundant layers to the MT task. In this section, we increase the level of granularity to that of attention heads instead of layers.

A well established finding for supervised encoder-decoder MT models, is that up to 90% of the attention heads can be pruned while minimising fall in translation performance (Voita et al., 2019; Behnke & Heafield, 2020; Michel et al., 2019). We note that asking about the extent of pruning is a slightly ill-formed research question, as it depends on the type of pruning technique used. However broad trends of highly prunable models have been observed in the supervised MT paradigm. In the in-context paradigm, there is no explicit supervision. Thus it is not clear if the task knowledge is spread across a much larger number of attention heads, or similarly specialised to a few heads; for instance, recently Bansal et al. (2023) studied attention-head importance for a broader set of ICL tasks, finding that the most important heads for ICL occur in the middle layers of the model.

### 6.1 Training Attention Head Gates with $L_0$ regularisation

For a scalable approach to pruning, we opt to train self-attention head gates following Voita et al. (2019) using the technique of differentiable $L_0$ regularization (Louizos et al., 2017). Let the attention head gates $g \in \mathbb{R}^{n_h \times n_\ell}$ be a set of trainable parameters, where $n_h$ is the number of attention heads per layer, and $n_\ell$ is the number of layers. Let the original output of each attention head be $v_j$, gated outputs $\tilde{v}_j$ are obtained by elementwise multiplication of the gate value $g_j$, i.e., $\tilde{v}_j = g_j \odot v_j$. For $\{(x, y)\}^n$ source sentence $(x)$ and target sentence $(y)$ training pairs, a model $f$ and loss function $\mathcal{L}$, $L_p$ regularisation adds a $\lambda$ weighted penalty associated with the complexity of the parameters. [6] The $L_0$ loss is non-differentiable as it involves raw counts of parameters. As a work around, $g$

---

[6] $L_2$ regularisation has the effect of reducing the magnitude of all $g$, $L_1$ regularisation has the effect of reducing the magnitude of several attention heads to a very small value (but not exactly 0), while $L_0$ regularisation has the effect of driving $g$ values to exactly 0.

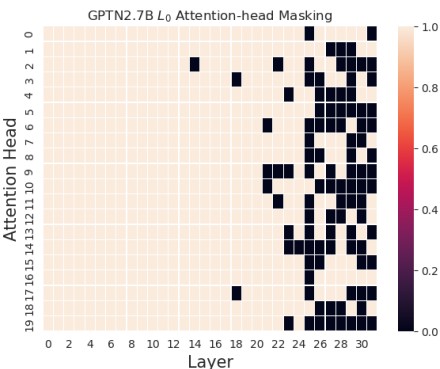 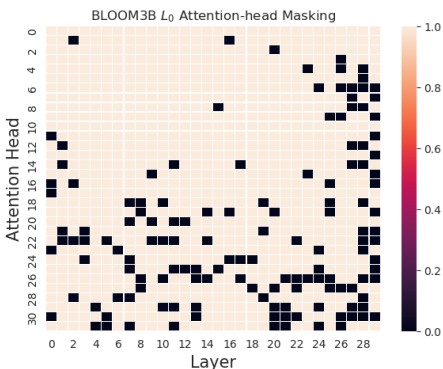

Figure 5: Visualisation of attention head masks for GPTNeo and BLOOM, learned with $L_0(\lambda = 0.01)$ regularisation under a `0-prompt train` scheme in en $\rightarrow$ fr. A value of 0 (in black) indicates that the attention head is effectively masked out by the trained attention gate. Around 10% of attention heads are masked out i.e., redundant, with a majority of them occuring at the later layers for GPTNeo and distributed across layers for BLOOM. fr $\rightarrow$ en is availble in Section A.9

can be approximated with random variables drawn from a Binary concrete distribution (Maddison et al., 2016; Jang et al., 2016).[7] We refer the reader to Louizos et al. (2017) for the relevant technical exposition. Details of training are provided in Section A.8.

## 6.2 STUDYING REDUNDANCY VIA COMPRESSION

Thus far, we have mainly focused on general patterns across the two models. However we noted that GPTNeo has some critical differences from BLOOM in terms of having critical layers (see Section 5.1), and having a steeper task recognition phase with greater redundancy towards the end layers (Section 4). Considering the redundancy, to what extent are there specialised attention heads for MT in the GPT-style models? If there were specialised heads, we would expect the model to be highly compressable/prunable to a select few heads. We plot a grid map of learned attention gate values for en $\rightarrow$ fr, where 0 indicates that the head is masked out (Figure 5). We find that most of the masked heads are distributed at the later layers for GPTNeo and are distributed across layers for BLOOM. This appears consistent with Section 4's and Section 5.1's observations that task-related layers are more easily isolated in GPTNeo and more distributed in BLOOM.

Although the pattern of distribution differs across the two models, we note that there are no "few" specialised heads, which directly contrasts with the literature on compression in supervised MT models (Voita et al., 2019; Michel et al., 2019). Potential reasons for this difference might include data distribution and model architecture, or cross-entropy loss associated with task tuning for MT. We leave this as an open question for future work.

## 7 CONCLUSION

We demonstrate evidence that GPT models locate, or learn, the translation task at a specific layer during in-context learning, which can be identified via causal masking of the context. Contrary to common fine-tuning wisdom, we show that it is sometimes beneficial to target middle layers of the model which could be associated with task recognition. Section 5.1 further corroborates this, by demonstrating that these *task location* layers most negatively impact performance when masked, while the later layers contain *task redundancy*. Finally, we trained attention head gates using differentiable $L_0$ regularisation (Section 6), and found that around 10% of attention heads can be masked. These are mostly distributed across the later layers of the model, suggesting that, similar to supervised NMT models, GPT models may contain a lot of redundant attention heads, but only in the layers of the model that are responsible for executing the translation task.

---

[7]The class of Concrete distributions was invented to work around the problem of automatic differentiation of stochastic computation graphs.

## 8   REPRODUCIBILITY

The MT dataset that we use, FLORES (Goyal et al., 2021), is fully open-source and well-known in the community. Models are open-source and freely available on Huggingface (Wolf et al., 2019). We used models of "reasonable" size (3B parameters) that can be run with consumer grade GPUs, making our reproducible to most academic institutions. Code to reproduce all the experiments will be made available subsequently.

## 9   ETHICAL CONCERNS

There are no known ethical concerns as these are exploratory studies on open-source LLMs. We also include a section on limitations at Appendix B.

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

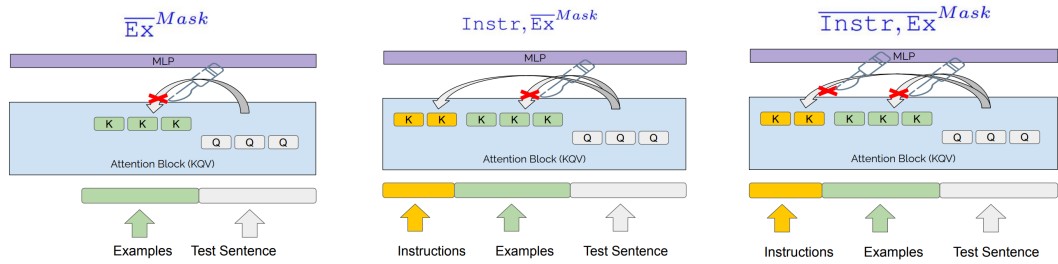

Figure 6: Graphical explanation of Masking the Attention over Instructions and Examples. The leftmost picture has no instructions and masks examples ($\overline{\text{Ex}}^{Mask}$), the middle picture has instructions and masks examples ($\text{Instr}, \overline{\text{Ex}}^{Mask}$), and the rightmost picture masks both instructions and examples ($\overline{\text{Instr, Ex}}^{Mask}$).

# A APPENDIX

## A.1 GRAPHICAL VIEW OF CONTEXT MASKING EXPERIMENTS

## A.2 EXAMPLE OF PROMPT FORMAT

| | | |
|---|---|---|
| Translate English to French. | | |
| English: A discomfort which lasts .. | French: | Un malaise qui dure |
| English: HTML is a language for formatting | French: | HTML est un langage de formatage |
| ... | | ... |
| English: After you become comfortable with formatting .. | French: | |

Table 3: A single continuous input sequence presented to the model for decoding a single test source sentence "After you become comfortable with formatting..". Given the entire sequence as input, the model proceeds to generate the target sequence.

## A.3 ADDITIONAL RESULTS ON ENGLISH & SPANISH

In addition to the language pairs en → fr and fr → en, we also run experiments on English and Spanish language pairs, both en → es and es → en. Due to space limitations, we plot the results of those experiments here. Overall, we see largely identical trends on both directions of English and Spanish to what we observe on Enlish and French translation tasks, leading us to conclude that our conclusions generalize across different translation tasks.

## A.4 ADDITIONAL RESULTS ON LLAMA7B-CHAT

We run context masking experiments on Llama7B-Chat Touvron et al. (2023) which is an instruction-tuned model. In contrast to GPTNeo and Bloom, the Llama model has no difficulty performing translation when the examples are masked, as long as instructions are present ($\text{Instr}\overline{\text{Ex}}^{Mask}$). However, when instructions are masked ($\overline{\text{Instr}\text{Ex}}^{Mask}$), the model suffers from catastrophic failure. This can be explained by the reliance on Instructions for an Instruction-tuned model (Figure 8).

## A.5 IS ATTENTION TO THE INPUT STILL NECESSARY?

One possible explanation for the results in Figure 1 is that, rather than identifying the point at which the task is recognized, we have identified the point at which the model no longer requires attending to *any* other tokens to encode it's input successfully. To explore this, we run experiments in the en → fr direction where we mask attention to *all inputs* from a certain layer onwards, rather than just the context; in this scenario, the representations of each token cannot attend to *any other token* past the specified.

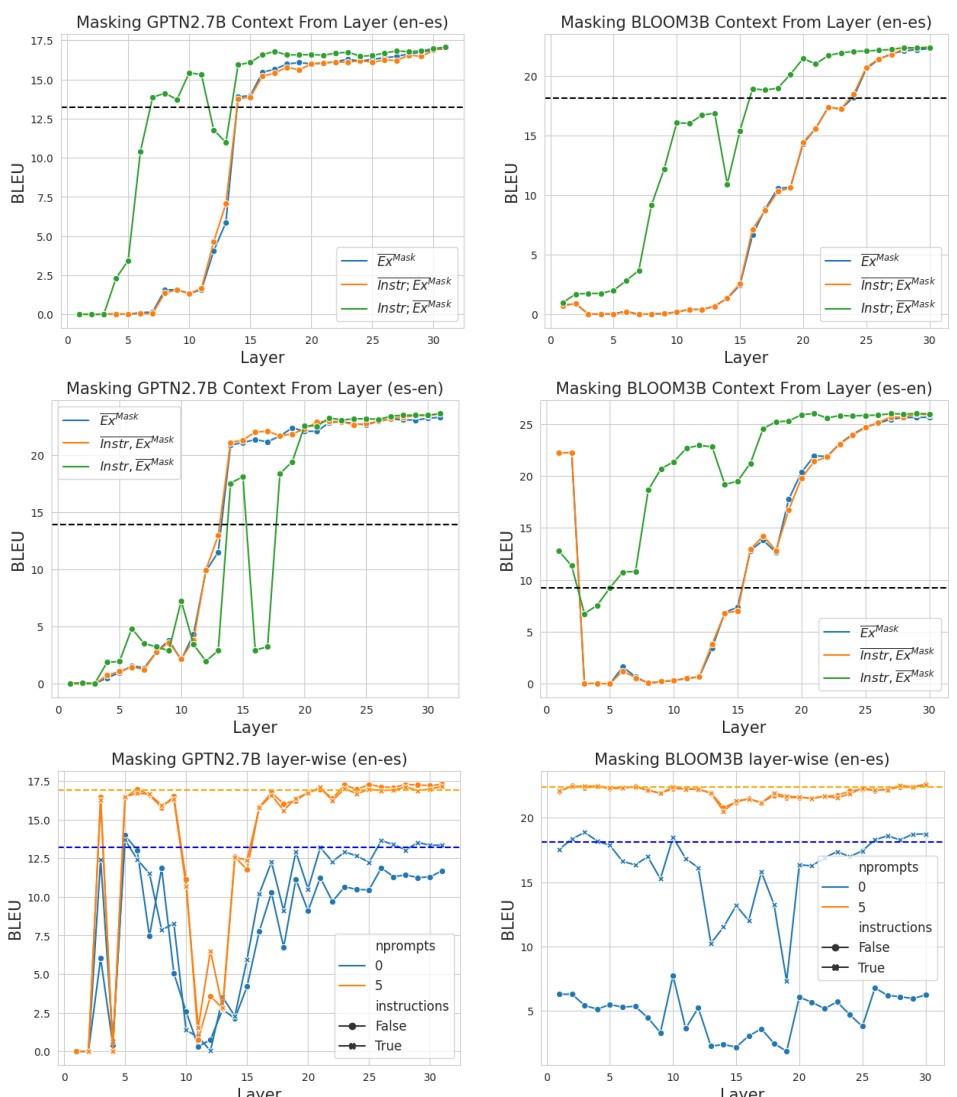

Figure 7: Context-masking and Layer-masking results on the **English to Spanish** translation task. Critically, we see nearly identical trends to what we see in Figure 1 and Figure 4 on the English to French translation task, suggesting our results generalize across language pairs.

We plot the results in Figure 9; we find that, for both GPTNeo and BLOOM, the layer at which attention can be fully removed is much higher than the layer at which we can remove attention to the context. In both models, removing attention before layer 20 results in catastrophic failure of the model, after which translation performance slowly improves; in GPTNeo, translation performance is never comparable to the baseline with no masking. Conversely, when masking only the context, translation performance improves as early as layer 10 and plateaus at the no-mask baseline much earlier. This suggests that the curves we observe in Figure 1 are due to the model having sufficiently attended to the context but still requiring attention to the input.

## A.6    AUTOREGRESSIVE DECODER ONLY TRANSFORMER

The transformer consists of stacked blocks of self-attention, which itself consists of smaller units of self-attention heads that are concatenated before being fed through a fully connected layer. In autoregressive decoder-only transformers, training and inference adopts a causal mask, where current positions are only able to attend to previous timesteps, instead of being able to attend to the entire

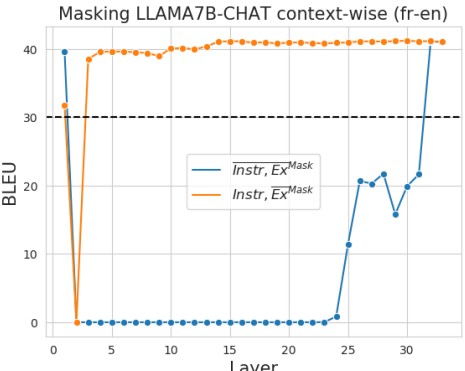
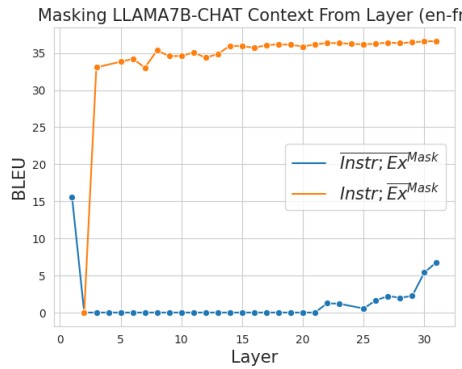

Figure 8: Context masking experiments for Instruction-tuned LLAMA7B on en ↔ fr when masking out from layer $j$ onwards.

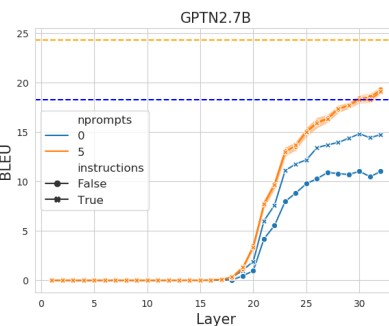
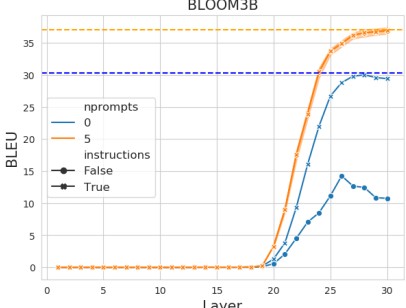

Figure 9: *Layer-from* experiments for GPTNEO2.7B and BLOOM3B on en → fr when masking out from layer $j$ onwards. Orange and blue dashed lines refer to the baselines (without masking) of 0 and 5 prompts with instructions.

input sequence. Unlike encoder-decoder NMT models where source and target sentence have separate processing transformer blocks, decoder-only means that the same model weights are both used to "encode" the source sentence and "decode" the target sentence in a single continuous sequence.

### A.7 TRAINING WITH AUTOREGRESSIVE TRANSLATION

The original language modeling objective in GPT training involves predicting the entire input token sequence which consists of both the source and target sentence (shifted by 1 position). We found this to produce slightly worse results than only minimising the negative log likelihood of predicting the target sentence to be translated, and not the entire sequence. We consider this autoregressive translation training.

### A.8 $L_0$ ATTENTION GATE TRAINING DETAILS

**Training Details** For Section 6.2, We train using Adam Optimizer ($\beta_1 = 0.9, \beta_2 = 0.999$) with a batch size of 32, and learning rate of 0.001, early stopping patience of 10 and threshold of 0.01. We initialise attention head gates to be 1 instead of random or 0.5 as this leads to faster convergence. We experiment with two different training settings, the `0-prompts Train` setting and the `5-prompts Train` setting. As described in Section A.7, we train the model by predicting only the target sentence, conditioned on the context. In the 0-prompt setting, the context consists of the instructions and the source sentence to be translated. In the 5-prompt setting, the context consists of the instructions, 5 prompt examples, and the source sentence to be translated.

In the `0-prompt` setting, the conditional prefix consists of the instructions and the source sentence to be translated. In the `5-prompt setting`, the conditional prefix consists of the instruction, 5 source target sentence pairs, and the source sentence to be translated.

**Data** We used the first 10,000 lines of en $\rightarrow$ fr from WMT06 Europarl Koehn (2005) for training.[8] To test the generalisability of trained attention head gates, we use a different test domain, FLORES Goyal et al. (2021) to reflect the scarcity of in-domain data. We also test an additional language direction en $\rightarrow$ pt in FLORES to see if training can generalise across languages.

**Training Details** We train using Adam Optimizer ($\beta_1 = 0.9, \beta_2 = 0.999$) with a batch size of 32, and learning rate of 0.001. We use a large early stopping patience of 10 and threshold of 0.01, and train for up to 100 epochs. This is due to the nature of $L_0$ training; we do not expect performance to improve over many iterations and would like the attention gates to keep training as long as there is no large loss in performance. We initialise attention head gates to be 1 instead of random or 0.5 as this leads to much faster convergence and better performance. For the regularisation weight $\lambda$, we search over a hyperparameter set of $\{0.1, 0.01, 0.001, 0.0001\}$ and found $0.01$ performs best on the validation set.

## A.9 $L_0$ HEAD MASKING EXPERIMENTS.

Additional experiments on L0 head masking in the fr$\rightarrow$ en and es$\rightarrow$ en direction.

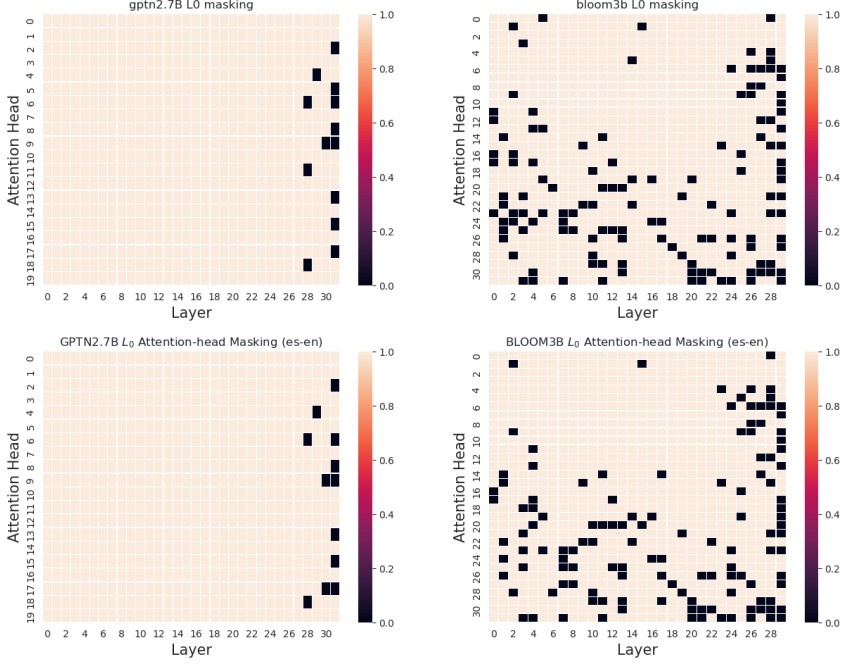

Figure 10: Visualisation of attention head masks for GPTNeo and BLOOM, learned with $L_0(\lambda = 0.01)$ regularisation under a `0-prompt train` scheme. A value of 0 (in black) indicates that the attention head is effectively masked out by the trained attention gate. A majority of them occuring at the later layers for GPTNeo and distributed across layers for BLOOM.

## A.10 GENERALISABILITY OF $L_0$ GATE TRAINING

We experiment with `0-prompts` and `5-prompts` in training and using $\lambda = 0$ (no regularisation) and $\lambda = 0.01$. $L_0$ training for the `0-prompts` shows some gains for the 0-prompts test case, and

---

[8]Data available from `https://www.statmt.org/europarl/`

| | Base | 0-prompts | | 5-prompts | | Base | 0-prompts | | 5-prompts | |
|---|---|---|---|---|---|---|---|---|---|---|
| | | $\lambda=0$ | $\lambda=.01$ | $\lambda=0$ | $\lambda=.01$ | | $\lambda=0$ | $\lambda=.01$ | $\lambda=0$ | $\lambda=.01$ |
| 0-prompts | 18.3 | 21.4 | 20.1 | 18.9 | 19.3 | 6.7 | 15.7 | 8.6 | 13.2 | 6.4 |
| 5-prompts | 24.3 | 24.5 | 24.1 | 23.6 | 24.2 | 25.9 | 19.6 | 25.8 | 24.3 | 26.0 |

| Train: en→fr, Test: en→fr | Train: en→fr, Test: en→pt |
|---|---|

Table 4: Performance when using trained attention head gates for $L_0$ with regularisation $\lambda = .01$. $\lambda = 0$ refers to training without regularisation. 0 and 5 prompts were used in the context for training. We highlight values which are greater or worse than 0.5 BLEU points from baseline. Note that as these are compression experiments, we do not expect $L_0$ to perform better than baseline.

with no loss on the 5-prompts test case (Table 4). Notably, this persists in en → pt, a different language direction from training.

The robustness of translation performance under multiple testing conditions (number of prompts, datasets, language directions) gives some confidence that the trained discrete attention head gates from $L_0$ support a general ability to translate (Table 4). In contrast, the soft attention head gates without regularisation ($\lambda = 0$) appear to overfit as they perform well on some conditions but deteriorate in others.

We observe that 0-prompt training for $L_0(\lambda = 0.01)$ also outperforms 5-prompts which is slightly suprising since 5-prompts has more information in the prefix to locate the translation task. One possibility is that the model overfit to the Europarl domain where the training prompts were drawn from.

### A.11 QUALITATIVE ANALYSIS OF LAYER-WISE MASKING

**GPTNEO** Masking $\ell_{4:8}$ results in a drop in performance for the 0-prompt setting but not the 5-prompt setting (Figure 4), which suggests that $\ell_{4:8}$ are **not** related to the processing of prompt examples. We emphasise that this interpretation mostly holds at an aggregate level and is not strictly for each instance. For Test Instance ID 575, the model still generates a copy of the English source sentence up to the masking of $\ell_{25}$ for the 0-prompts without instructions setting (Table 6). This suggests that uncertainty over the task is maintained across layers even though the contributions towards *task location* may be greater from specific layers.

**BLOOM** is observed to be more robust to masking of layers; suggesting that task location is more distributed. For the 5-prompt setting, the performance only decreases very slightly. For the 0-prompt setting, we observe that similar to GPTNEO, performance drops when masking out the middle layers. At the aggregate level, BLOOM appears to still be translating ($> 0$ BLEU) even when layers are masked. However we observe that the drop in performance is due to around 40 to 50% of the test sentences scoring $< 5$ BLEU points. There is a clear failure to translate, not simply producing poorer translations.

| layer | id | lang | BLEU | text |
|---|---|---|---|---|
| 1 | 600 | cy | 0.00 | uffose |
| 1 | 575 | ca | 0.00 | B marriages{ |
| 2 | 600 | et | 0.00 | sses room ( I |
| 2 | 575 | no | 0.00 | NaN |
| 3 | 600 | fr | 1.90 | C'est la même chose que l'on a fait avec les virus. |
| 3 | 575 | fr | 88.44 | Mais il y a beaucoup de choses sur les oiseaux qui ressemblent encore à un dinosaur. |
| 4 | 600 | no | 0.00 | NaN |
| 4 | 575 | no | 0.00 | NaN |
| 5 | 600 | fr | 78.78 | Hershey et Chase ont utilisé des phages, ou des virus, pour implanter leur propre gène dans un bactérie. |
| 5 | 575 | fr | 72.98 | Mais il y a beaucoup de choses sur les oiseaux qui ressemblent toujours à un dinosaur. |
| 6 | 600 | fr | 78.78 | Hershey et Chase ont utilisé des phages, ou des virus, pour implanter leur propre gène dans un bactérie. |
| 6 | 575 | fr | 60.29 | Mais il y a beaucoup de choses à propos de oiseaux qui ressemblent encore à un dinosaur. |
| 7 | 600 | fr | 13.94 | Hershey et Chase ont implanté leur propre gène dans un bactérie. |
| 7 | 575 | fr | 72.98 | Mais il y a beaucoup de choses sur les oiseaux qui ressemblent toujours à un dinosaur. |
| 8 | 600 | no | 0.00 | NaN |
| 8 | 575 | fr | 88.44 | Mais il y a beaucoup de choses sur les oiseaux qui ressemblent encore à un dinosau. |
| 9 | 600 | no | 0.00 | NaN |
| 9 | 575 | fr | 82.82 | Mais il y a beaucoup de choses sur les oiseaux qui ressemblent toujours à un dinosaure. |
| 10 | 600 | no | 0.00 | NaN |
| 10 | 575 | en | 2.73 | But there are a lot of things about birds that still look like a dinosaur. |
| 11 | 600 | en | 4.78 | Hershey and Chase used phages, or viruses, to implant their own DNA into a bacterium. |
| 11 | 575 | en | 2.73 | But there are a lot of things about birds that still look like a dinosaur. |
| 12 | 600 | no | 0.00 | NaN |
| 12 | 575 | no | 0.00 | NaN |
| 13 | 600 | no | 0.00 | NaN |
| 13 | 575 | fr | 35.75 | Mais il y a beaucoup de choses que je ne comprends pas. |
| 14 | 600 | en | 4.78 | Hershey and Chase used phages, or viruses, to implant their own DNA into a bacterium. |
| 14 | 575 | en | 2.73 | But there are a lot of things about birds that still look like a dinosaur. |
| 15 | 600 | fr | 76.48 | Hershey et Chase ont utilisé des phages, ou des virus, pour implanter leur propre gène dans un bacillus. |
| 15 | 575 | en | 2.73 | But there are a lot of things about birds that still look like a dinosaur. |
| 16 | 600 | fr | 78.78 | Hershey et Chase ont utilisé des phages, ou des virus, pour implanter leur propre gène dans un bactérie. |
| 16 | 575 | fr | 70.18 | Mais il y a beaucoup de choses sur les oiseaux qui ressemblent toujours comme un dinosaurof. |
| 17 | 600 | fr | 82.32 | Les Hershey et Chase ont utilisé des phages, ou des virus, pour implanter leur propre gène dans une bactérie. |
| 17 | 575 | fr | 88.44 | Mais il y a beaucoup de choses sur les oiseaux qui ressemblent encore à un dinosaur. |
| 18 | 600 | fr | 78.78 | Hershey et Chase ont utilisé des phages, ou des virus, pour implanter leur propre génome dans un bactérie. |
| 18 | 575 | fr | 66.38 | Mais il y a beaucoup de choses sur les oiseaux qui aussi ressemble à un dinosaures. |
| 19 | 600 | fr | 59.33 | Les héritiers de Hershey et de Chase ont utilisé des phages, ou des virus, pour implanter leur propre gène dans un bactérie. |
| 19 | 575 | fr | 47.91 | Mais il y a beaucoup de choses à propos de les oiseaux qui ressemblent toujours à un dinosaur. |
| 20 | 600 | fr | 48.82 | Hershey et Chase ont utilisé les phages, ou les virus, pour implanter leur propre gène dans un bactérie. |
| 20 | 575 | en | 2.73 | But there are a lot of things about birds that still look like a dinosaur. |
| 21 | 600 | fr | 78.78 | Hershey et Chase ont utilisé des phages, ou des virus, pour implanter leur propre gène dans un bactérie. |
| 21 | 575 | fr | 88.44 | Mais il y a beaucoup de choses sur les oiseaux qui ressemblent encore à un dinosaur. |
| 22 | 600 | fr | 48.82 | Hershey et Chase ont utilisé les phages, ou les virus, pour implanter leur propre gène dans un bactérie. |
| 22 | 575 | fr | 88.44 | Mais il y a beaucoup de choses sur les oiseaux qui ressemblent encore à un dinosaurof. |
| 23 | 600 | fr | 78.78 | Hershey et Chase ont utilisé des phages, ou des virus, pour implanter leur propre gène dans un bactérie. |
| 23 | 575 | fr | 88.44 | Mais il y a beaucoup de choses sur les oiseaux qui ressemblent encore à un dinosaur. |
| 24 | 600 | fr | 78.78 | Hershey et Chase ont utilisé des phages, ou des virus, pour implanter leur propre génome dans un bactérie. |
| 24 | 575 | fr | 62.72 | Mais il y a beaucoup de choses à propos de les oiseaux qui ressemblent encore à un dinosaur. |
| 25 | 600 | fr | 78.78 | Hershey et Chase ont utilisé des phages, ou des virus, pour implanter leur propre gène dans un bactérie. |
| 25 | 575 | fr | 88.44 | Mais il y a beaucoup de choses sur les oiseaux qui ressemblent encore à un dinosaur. |
| 26 | 600 | fr | 78.78 | Hershey et Chase ont utilisé des phages, ou des virus, pour implanter leur propre gène dans un bactérie. |
| 26 | 575 | fr | 88.44 | Mais il y a beaucoup de choses sur les oiseaux qui ressemblent encore à un dinosaur. |
| 27 | 600 | fr | 66.28 | Hershey et Château ont utilisé des phages, ou des virus, pour implanter leur propre gène dans un bactérie. |
| 27 | 575 | fr | 88.44 | Mais il y a beaucoup de choses sur les oiseaux qui ressemblent encore à un dinosaur. |
| 28 | 600 | fr | 78.78 | Hershey et Chase ont utilisé des phages, ou des virus, pour implanter leur propre gène dans un bactérie. |
| 28 | 575 | fr | 88.44 | Mais il y a beaucoup de choses sur les oiseaux qui ressemblent encore à un dinosaur. |
| 29 | 600 | fr | 59.33 | Les héritiers de Hershey et de Chase ont utilisé des phages, ou des virus, pour implanter leur propre gène dans un bactérie. |
| 29 | 575 | fr | 88.44 | Mais il y a beaucoup de choses sur les oiseaux qui ressemblent encore à un dinosaurof. |
| 30 | 600 | fr | 78.78 | Hershey et Chase ont utilisé des phages, ou des virus, pour implanter leur propre gène dans un bactérie. |
| 30 | 575 | fr | 88.44 | Mais il y a beaucoup de choses sur les oiseaux qui ressemblent encore à un dinosaur. |
| 31 | 600 | fr | 78.78 | Hershey et Chase ont utilisé des phages, ou des virus, pour implanter leur propre gène dans un bactérie. |
| 31 | 575 | fr | 88.44 | Mais il y a beaucoup de choses sur les oiseaux qui ressemblent encore à un dinosaur. |
| 32 | 600 | fr | 6.44 | Les héritiers de Hershey et de Chase ont été capables de l'implanter dans un bactérie. |
| 32 | 575 | fr | 51.52 | Mais il y a beaucoup de choses sur les oiseaux que sont encore aussi vus comme un dinosaures. |

Table 5: 0-prompts with instructions, masking layer by layer of GPTNEO2.7B

| layer | id | lang | BLEU | text |
|---|---|---|---|---|
| 1 | 575 | no | 0.0 | NaN |
| 1 | 600 | no | 0.0 | NaN |
| 2 | 575 | NaN | 0.0 | , |
| 2 | 600 | no | 0.0 | NaN |
| 3 | 575 | fr | 2.3 | C'est pas un oiseau, c'est un dinosaur. |
| 3 | 600 | fr | 0.8 | [phare] Phare, phare, phare, phare, phare, phare, phare, phare, phare, phare, phare, ..., |
| 4 | 575 | en | 2.7 | "I think it's a dinosaur, I think it's a dinosaur." |
| 4 | 600 | fr | 2.6 | Les virus, c'est-ce qu'on dit? C'un mot? C'est pas un mot? C'un mot? C'un'un? ... |
| 5 | 575 | fr | 42.9 | Mais il y a beaucoup de choses à propos de oiseaux qui ressemblent toujours comme un dinosaur. |
| 5 | 600 | fr | 73.6 | L'Hershey et Chase ont utilisé des phages, ou des virus, pour implanter leur propre gène dans un bactérie. |
| 6 | 575 | fr | 53.8 | Mais il y a beaucoup de choses à propos de oiseaux qui ressemblent toujours à un dinosaure. |
| 6 | 600 | fr | 74.9 | Les Hershey et Chase ont utilisé des phages, ou des virus, pour implanter leur propre gène dans un bactérie. |
| 7 | 575 | fr | 76.2 | Et il y a beaucoup de choses sur les oiseaux qui ressemblent toujours à un dinosaure. |
| 7 | 600 | fr | 83.8 | Les Hershey et Chase ont utilisé des phages, ou des virus, pour implanter leur propre ADN dans un bactérie. |
| 8 | 575 | fr | 88.4 | Mais il y a beaucoup de choses sur les oiseaux qui ressemblent encore à un dinosau. |
| 8 | 600 | fr | 83.8 | L'usine Hershey et Chase ont utilisé des phages, ou des virus, pour implanter leur propre ADN dans un bactérie. |
| 9 | 575 | en | 1.5 | The bird is a dinosaur. |
| 9 | 600 | fr | 33.9 | Les hémoglobine et Chase utilisent les phages, ou les virus, pour implanter leur propre gène dans un bactérie. |
| 10 | 575 | en | 2.7 | But there are a lot of things about birds that still look like a dinosaur. |
| 10 | 600 | fr | 11.4 | Les phages, ou virus, ont implanté leur propre gène dans un bactérie. |
| 11 | 575 | en | 2.6 | I think it's a good idea to have a little bit of a bird in your pocket. |
| 11 | 600 | en | 0.0 | The French have a saying: "The French have a saying: "The French have a saying: "The French have a saying:... |
| 12 | 575 | en | 2.7 | But there are a lot of things about birds that still look like a dinosaur. |
| 12 | 600 | en | 1.7 | The bacterium was then able to use the phage to infect other bacteria. |
| 13 | 575 | en | 2.7 | But there are a lot of things about birds that still look like a dinosaur. |
| 13 | 600 | fr | 18.7 | L'entreprise Hershey a utilisé des phages pour implanter leur propre DNA dans leur bactérie. |
| 14 | 575 | en | 2.7 | But there are a lot of things about birds that still look like a dinosaur. |
| 14 | 600 | en | 4.8 | Hershey and Chase used phages, or viruses, to implant their own DNA into a bacterium. |
| 15 | 575 | fr | 3.0 | C'est pas un truc de poulet, c'est un truc de poulet. |
| 15 | 600 | fr | 35.7 | L'université de Paris-Sud a utilisé des phages, ou viraux, pour implanter leur propre gène dans un bacillus. |
| 16 | 575 | en | 2.7 | But there are a lot of things about birds that still look like a dinosaur. |
| 16 | 600 | fr | 74.9 | Les Hershey et Chase ont utilisé des phages, ou des virus, pour implanter leur propre génome dans un bactérie. |
| 17 | 575 | en | 2.7 | But there are a lot of things about birds that still look like a dinosaur. |
| 17 | 600 | fr | 82.3 | Les Hershey et Chase ont utilisé des phages, ou des virus, pour implanter leur propre gène dans une bactérie. |
| 18 | 575 | en | 2.7 | But there are a lot of things about birds that still look like a dinosaur. |
| 18 | 600 | fr | 74.9 | Les Hershey et Chase ont utilisé des phages, ou des virus, pour implanter leur propre génome dans un bactérie. |
| 19 | 575 | fr | 44.2 | Mais il y a beaucoup de choses à propos de oiseaux qui ressemblent toujours à un dinosaur. |
| 19 | 600 | fr | 59.3 | Les héritiers de Hershey et de Chase ont utilisé des phages, ou des virus, pour implanter leur propre gène dans un bactérie. |
| 20 | 575 | en | 2.7 | But there are a lot of things about birds that still look like a dinosaur. |
| 20 | 600 | fr | 46.4 | Les Hershey et Chase ont utilisé les phages, ou les virus, pour implanter leur propre gène dans un bactérie. |
| 21 | 575 | en | 2.7 | But there are a lot of things about birds that still look like a dinosaur. |
| 21 | 600 | fr | 74.9 | L'usine Hershey et Chase ont utilisé des phages, ou des virus, pour implanter leur propre gène dans un bactérie. |
| 22 | 575 | en | 2.7 | But there are a lot of things about birds that still look like a dinosaur. |
| 22 | 600 | fr | 56.3 | Les Hershey et Chase ont utilisé les phages, ou les virus, pour implanter leur propre ADN dans un bactérie. |
| 23 | 575 | en | 2.7 | But there are a lot of things about birds that still look like a dinosaur. |
| 23 | 600 | fr | 82.9 | L'Hershey et Chase ont utilisé des phages, ou des virus, pour implanter leur propre ADN dans un bactérie. |
| 24 | 575 | fr | 60.3 | Mais il y a beaucoup de choses à propos de oiseaux qui ressemblent encore à un dinosaur. |
| 24 | 600 | fr | 37.0 | L'usine Hershey et Chase utilisaient les phages, ou les virus, pour implanter leur propre génome dans un bactérie. |
| 25 | 575 | en | 2.7 | But there are a lot of things about birds that still look like a dinosaur. |
| 25 | 600 | fr | 74.9 | Les Hershey et Chase ont utilisé des phages, ou des virus, pour implanter leur propre gène dans un bactérie. |
| 26 | 575 | fr | 71.6 | Mais il y a beaucoup de choses à propos de oiseaux qui ressemblent encore à un dinosaure. |
| 26 | 600 | fr | 73.6 | L'Hershey et Chase ont utilisé des phages, ou des virus, pour implanter leur propre gène dans un bactérie. |
| 27 | 575 | fr | 88.4 | Mais il y a beaucoup de choses sur les oiseaux qui ressemblent encore à un dinosaurof. |
| 27 | 600 | fr | 63.0 | Les Hershey et Château ont utilisé des phages, ou des virus, pour implanter leur propre gène dans un bactérie. |
| 28 | 575 | fr | 44.2 | Mais il y a beaucoup de choses à propos de oiseaux qui ressemblent toujours à un dinosaur. |
| 28 | 600 | fr | 74.9 | Les Hershey et Chase ont utilisé des phages, ou des virus, pour implanter leur propre gène dans un bactérie. |
| 29 | 575 | fr | 87.0 | Mais il y a beaucoup de choses sur les oiseaux qui ressemblent encore à un dinosaurofrench: |
| 29 | 600 | fr | 53.4 | L'entreprise de la filière Hershey a utilisé des phages, ou des virus, pour implanter leur propre gène dans un bactérie. |
| 30 | 575 | fr | 82.8 | Mais il y a beaucoup de choses sur les oiseaux qui ressemblent toujours à un dinosaure. |
| 30 | 600 | fr | 74.9 | Les Hershey et Chase ont utilisé des phages, ou des virus, pour implanter leur propre gène dans un bactérie. |
| 31 | 575 | fr | 82.8 | Mais il y a beaucoup de choses sur les oiseaux qui ressemblent toujours à un dinosaure. |
| 31 | 600 | fr | 59.3 | Les hémoglobins de Hershey et de Chase ont utilisé des phages, ou des virus, pour implanter leur propre gène dans un bactérie. |
| 32 | 575 | fr | 67.5 | Mais il y a beaucoup de choses sur les oiseaux qui aussi ressemblent à un dinosaurof. |
| 32 | 600 | fr | 6.7 | L'Hershey et le Chase ont été capables de l'implanter leur propre gène dans un bactérie. |

Table 6: 0-prompts without instructions, masking layer by layer of GPTNEO2.7B

| layer | id | lang | BLEU | text |
|---|---|---|---|---|
| 1 | 902 | en | 0.0 | : of |
| 1 | 575 | en | 0.0 | of |
| 2 | 902 | en | 0.0 | of(n, very very- of  S First |
| 2 | 575 | da | 0.0 | f( |
| 3 | 902 | fr | 100.0 | Les scènes sont affichées sur les pyramides et les différentes pyramides sont éclairées. |
| 3 | 575 | fr | 88.4 | Mais il y a beaucoup de choses sur les oiseaux qui ressemblent encore à un dinosaur. |
| 4 | 902 | en | 0.0 | the, the French, the French, the English, the, the, the, the, the, the, the, the, the, the, |
| 4 | 575 | no | 0.0 | NaN |
| 5 | 902 | fr | 65.9 | Les scènes sont affichées sur les pyramides et les différents pyramides sont éclairés. |
| 5 | 575 | fr | 88.4 | Mais il y a beaucoup de choses sur les oiseaux qui ressemblent encore à un dinosaur. |
| 6 | 902 | fr | 100.0 | Les scènes sont affichées sur les pyramides et les différentes pyramides sont éclairées. |
| 6 | 575 | fr | 88.4 | Mais il y a beaucoup de choses sur les oiseaux qui ressemblent encore à un dinosaur. |
| 7 | 902 | fr | 100.0 | Les scènes sont affichées sur les pyramides et les différentes pyramides sont éclairées. |
| 7 | 575 | fr | 100.0 | Mais il y a beaucoup de choses sur les oiseaux qui ressemblent encore à un dinosaure. |
| 8 | 902 | fr | 100.0 | Les scènes sont affichées sur les pyramides et les différentes pyramides sont éclairées. |
| 8 | 575 | fr | 100.0 | Mais il y a beaucoup de choses sur les oiseaux qui ressemblent encore à un dinosaure. |
| 9 | 902 | fr | 65.9 | Les scènes sont affichées sur les pyramides et les différents pyramides sont éclairés. |
| 9 | 575 | fr | 100.0 | Mais il y a beaucoup de choses sur les oiseaux qui ressemblent encore à un dinosaure. |
| 10 | 902 | fr | 100.0 | Les scènes sont affichées sur les pyramides et les différentes pyramides sont éclairées. |
| 10 | 575 | fr | 100.0 | Mais il y a beaucoup de choses sur les oiseaux qui ressemblent encore à un dinosaure. |
| 11 | 902 | fr | 1.4 | Fondamentalement, vous afficherez des annonces pour proposer votre aide, arpenterez les quais, ... |
| 11 | 575 | fr | 1.3 | Fondamentalement, vous afficherez des annonces pour proposer votre aide, arpenterez les quais, ... |
| 12 | 902 | fr | 100.0 | Les scènes sont affichées sur les pyramides et les différentes pyramides sont éclairées. |
| 12 | 575 | fr | 42.5 | Mais il y a beaucoup de choses qui ressemblent à un dinosaur. |
| 13 | 902 | fr | 34.5 | Les scènes sont déclarées sur les pyramides et les pyramides sont déclarées sur les pyramides. |
| 13 | 575 | fr | 5.5 | Les oiseaux sont des animaux, mais ils sont aussi des êtres humains. |
| 14 | 902 | fr | 100.0 | Les scènes sont affichées sur les pyramides et les différentes pyramides sont éclairées. |
| 14 | 575 | fr | 73.3 | Mais il y a beaucoup de choses sur les oiseaux qui ressemblent à un dinosauresque. |
| 15 | 902 | fr | 64.0 | Les scènes sont affichées sur les pyramides et les pyramides différents sont éclairés. |
| 15 | 575 | fr | 26.7 | Mais il y a beaucoup de choses à propos de la façon dont les oiseaux se ressemblent, même si c'est un peu plus tard. |
| 16 | 902 | fr | 100.0 | Les scènes sont affichées sur les pyramides et les différentes pyramides sont éclairées. |
| 16 | 575 | fr | 76.7 | Mais il y a beaucoup de choses sur les oiseaux qui ressemblent encore comme un dinosaures. |
| 17 | 902 | fr | 100.0 | Les scènes sont affichées sur les pyramides et les différentes pyramides sont éclairées. |
| 17 | 575 | fr | 88.4 | Mais il y a beaucoup de choses sur les oiseaux qui ressemblent encore à un dinosaur. |
| 18 | 902 | fr | 65.9 | Les scènes sont affichées sur les pyramides et les différents pyramides sont éclairés. |
| 18 | 575 | fr | 88.4 | Mais il y a beaucoup de choses sur les oiseaux qui ressemblent encore à un dinosauresque. |
| 19 | 902 | fr | 65.9 | Les scènes sont affichées sur les pyramides et les différents pyramides sont illuminés. |
| 19 | 575 | fr | 88.4 | Mais il y a beaucoup de choses sur les oiseaux qui ressemblent encore à un dinosaur. |
| 20 | 902 | fr | 65.9 | Les scènes sont affichées sur les pyramides et les différents pyramides sont éclairés. |
| 20 | 575 | fr | 88.4 | Mais il y a beaucoup de choses sur les oiseaux qui ressemblent encore à un dinosaur. |
| 21 | 902 | fr | 100.0 | Les scènes sont affichées sur les pyramides et les différentes pyramides sont éclairées. |
| 21 | 575 | fr | 88.4 | Mais il y a beaucoup de choses sur les oiseaux qui ressemblent encore à un dinosaur. |
| 22 | 902 | fr | 64.0 | Les scènes sont affichées sur les pyramides et les pyramides différents sont éclairés. |
| 22 | 575 | fr | 88.4 | Mais il y a beaucoup de choses sur les oiseaux qui ressemblent encore à un dinosauroide. |
| 23 | 902 | fr | 65.9 | Les scènes sont affichées sur les pyramides et les différents pyramides sont éclairés. |
| 23 | 575 | fr | 88.4 | Mais il y a beaucoup de choses sur les oiseaux qui ressemblent encore à un dinosaur. |
| 24 | 902 | fr | 78.3 | Les scènes sont affichées sur les pyramides et les différents pyramides sont éclairées. |
| 24 | 575 | fr | 88.4 | Mais il y a beaucoup de choses sur les oiseaux qui ressemblent encore à un dinosaur. |
| 25 | 902 | fr | 100.0 | Les scènes sont affichées sur les pyramides et les différentes pyramides sont éclairées. |
| 25 | 575 | fr | 88.4 | Mais il y a beaucoup de choses sur les oiseaux qui ressemblent encore à un dinosaur. |
| 26 | 902 | fr | 65.9 | Les scènes sont affichées sur les pyramides et les différents pyramides sont éclairés. |
| 26 | 575 | fr | 88.4 | Mais il y a beaucoup de choses sur les oiseaux qui ressemblent encore à un dinosaur. |
| 27 | 902 | fr | 100.0 | Les scènes sont affichées sur les pyramides et les différentes pyramides sont éclairées. |
| 27 | 575 | fr | 88.4 | Mais il y a beaucoup de choses sur les oiseaux qui ressemblent encore à un dinosaur. |
| 28 | 902 | fr | 65.9 | Les scènes sont affichées sur les pyramides et les différents pyramides sont éclairés. |
| 28 | 575 | fr | 88.4 | Mais il y a beaucoup de choses sur les oiseaux qui ressemblent encore à un dinosaur. |
| 29 | 902 | fr | 65.9 | Les scènes sont affichées sur les pyramides et les différents pyramides sont éclairés. |
| 29 | 575 | fr | 100.0 | Mais il y a beaucoup de choses sur les oiseaux qui ressemblent encore à un dinosaure. |
| 30 | 902 | fr | 78.3 | Les scènes sont affichées sur les pyramides et les différents pyramides sont éclairées. |
| 30 | 575 | fr | 88.4 | Mais il y a beaucoup de choses sur les oiseaux qui ressemblent encore à un dinosaur. |
| 31 | 902 | fr | 65.9 | Les scènes sont affichées sur les pyramides et les différents pyramides sont éclairés. |
| 31 | 575 | fr | 88.4 | Mais il y a beaucoup de choses sur les oiseaux qui ressemblent encore à un dinosaur. |
| 32 | 902 | fr | 65.9 | Les scènes sont affichées sur les pyramides et les différents pyramides sont illuminées. |
| 32 | 575 | fr | 67.5 | Mais il y a beaucoup de choses sur les oiseaux qui aujourd'hui ressemblent à un dinosauro. |

Table 7: 5-prompts with instructions, masking layer by layer of GPTNEO2.7B

| layer | id | lang | BLEU | text |
|---|---|---|---|---|
| 1 | 600 | tl | 0.0 | *- ing |
| 1 | 575 | en | 0.6 | fl.of, |
| 2 | 600 | en | 1.6 | in " " - ( –, -, - (es," " " so " whats " whats" " between whats –what e, |
| 2 | 575 | en | 2.3 | " ",what awaited ico " " " "_, . |
| 3 | 600 | fr | 86.6 | Hershey et Chase ont utilisé des phages, ou des virus, pour implanter leur propre gène dans une bactérie. |
| 3 | 575 | fr | 88.4 | Mais il y a beaucoup de choses sur les oiseaux qui ressemblent encore à un dinosaur. |
| 4 | 600 | en | 0.3 | the, etc. |
| 4 | 575 | ro | 0.0 | are: are: are: are: are: are: are: are: are: are: are: are: are: are: are: are: are: ... |
| 5 | 600 | fr | 76.5 | Hershey et Chase ont utilisé des phages, ou des virus, pour implanter leur propre gène dans un bacille. |
| 5 | 575 | fr | 88.4 | Mais il y a beaucoup de choses sur les oiseaux qui ressemblent encore à un dinosaur. |
| 6 | 600 | fr | 88.1 | Hershey et Chase ont utilisé des phages, ou des virus, pour implanter leur propre ADN dans un bactérie. |
| 6 | 575 | fr | 62.7 | Mais il y a beaucoup de choses à propos de les oiseaux qui ressemblent encore à un dinosaur. |
| 7 | 600 | fr | 88.1 | Hershey et Chase ont utilisé des phages, ou des virus, pour implanter leur propre ADN dans un bactérie. |
| 7 | 575 | fr | 100.0 | Mais il y a beaucoup de choses sur les oiseaux qui ressemblent encore à un dinosaure. |
| 8 | 600 | fr | 85.7 | Hershey et Chase ont utilisé des phages, ou des virus, pour implanter leur propre ADN dans un bacillus. |
| 8 | 575 | fr | 100.0 | Mais il y a beaucoup de choses sur les oiseaux qui ressemblent encore à un dinosaure. |
| 9 | 600 | fr | 78.8 | Hershey et Chase ont utilisé des phages, ou des virus, pour implanter leur propre gène dans un bactérie. |
| 9 | 575 | fr | 100.0 | Mais il y a beaucoup de choses sur les oiseaux qui ressemblent encore à un dinosaure. |
| 10 | 600 | fr | 30.1 | En gros, vous mettre en place des phages, ou viraux, pour implanter leur propre gène dans un bactérie. |
| 10 | 575 | fr | 100.0 | Mais il y a beaucoup de choses sur les oiseaux qui ressemblent encore à un dinosaure. |
| 11 | 600 | fr | 1.3 | Fondamentalement, vous afficherez des annonces pour proposer votre aide, arpenterez les quais, aborderez les personnes nettoyant leurs yachts, ... |
| 11 | 575 | fr | 1.4 | Fondamentalement, vous afficherez des annonces pour proposer votre aide, arpenterez les quais, aborderez les personnes nettoyant leurs yachts, . |
| 12 | 600 | fr | 12.4 | Les phages sont utilisés pour implanter leur propre gène dans un virus. |
| 12 | 575 | fr | 36.8 | Mais il y a beaucoup de choses qui semblent être des dinosaures. |
| 13 | 600 | fr | 4.3 | Les phages sont des virus qui sont implantés dans la cellule d'un organisme. |
| 13 | 575 | fr | 5.5 | Les oiseaux sont des animaux, mais ils sont aussi des êtres humains. |
| 14 | 600 | fr | 74.4 | Hershey et Chase ont utilisé des phages, ou virus, pour implanter leur propre ADN dans un bactérie. |
| 14 | 575 | fr | 73.3 | Mais il y a beaucoup de choses sur les oiseaux qui ressemblent à un dinosauresque. |
| 15 | 600 | fr | 62.6 | Hershey et Chase ont utilisé des phages, ou virus, pour implanter leur propre gène dans un bacille. |
| 15 | 575 | fr | 26.7 | Mais il y a beaucoup de choses à propos de la façon dont les oiseaux se ressemblent, même si c'est un peu plus tard. |
| 16 | 600 | fr | 85.7 | Hershey et Chase ont utilisé des phages, ou des virus, pour implanter leur propre ADN dans un bacillus. |
| 16 | 575 | fr | 76.7 | Mais il y a beaucoup de choses sur les oiseaux qui ressemblent encore comme un dinosaures. |
| 17 | 600 | fr | 85.7 | Hershey et Chase ont utilisé des phages, ou des virus, pour implanter leur propre ADN dans un bacille. |
| 17 | 575 | fr | 88.4 | Mais il y a beaucoup de choses sur les oiseaux qui ressemblent encore à un dinosauresque. |
| 18 | 600 | fr | 85.7 | Hershey et Chase ont utilisé des phages, ou des virus, pour implanter leur propre ADN dans un bacille. |
| 18 | 575 | fr | 88.4 | Mais il y a beaucoup de choses sur les oiseaux qui ressemblent encore à un dinosauresque. |
| 19 | 600 | fr | 76.5 | Hershey et Chase ont utilisé des phages, ou des virus, pour implanter leur propre gène dans un bacille. |
| 19 | 575 | fr | 88.4 | Mais il y a beaucoup de choses sur les oiseaux qui ressemblent encore à un dinosaur. |
| 20 | 600 | fr | 76.5 | Hershey et Chase ont utilisé des phages, ou des virus, pour implanter leur propre gène dans un bacillus. |
| 20 | 575 | fr | 88.4 | Mais il y a beaucoup de choses sur les oiseaux qui ressemblent encore à un dinosaur. |
| 21 | 600 | fr | 88.1 | Hershey et Chase ont utilisé des phages, ou des virus, pour implanter leur propre ADN dans un bactérie. |
| 21 | 575 | fr | 88.4 | Mais il y a beaucoup de choses sur les oiseaux qui ressemblent encore à un dinosaur. |
| 22 | 600 | fr | 85.7 | Hershey et Chase ont utilisé des phages, ou des virus, pour implanter leur propre ADN dans un bacille. |
| 22 | 575 | fr | 88.4 | Mais il y a beaucoup de choses sur les oiseaux qui ressemblent encore à un dinosauro. |
| 23 | 600 | fr | 88.1 | Hershey et Chase ont utilisé des phages, ou des virus, pour implanter leur propre ADN dans un bactérie. |
| 23 | 575 | fr | 88.4 | Mais il y a beaucoup de choses sur les oiseaux qui ressemblent encore à un dinosaur. |
| 24 | 600 | fr | 85.7 | Hershey et Chase ont utilisé des phages, ou des virus, pour implanter leur propre ADN dans un bacille. |
| 24 | 575 | fr | 88.4 | Mais il y a beaucoup de choses sur les oiseaux qui ressemblent encore à un dinosaur. |
| 25 | 600 | fr | 76.5 | Hershey et Chase ont utilisé des phages, ou des virus, pour implanter leur propre gène dans un bacillus. |
| 25 | 575 | fr | 88.4 | Mais il y a beaucoup de choses sur les oiseaux qui ressemblent encore à un dinosaur. |
| 26 | 600 | fr | 76.5 | Hershey et Chase ont utilisé des phages, ou des virus, pour implanter leur propre gène dans un bacille. |
| 26 | 575 | fr | 88.4 | Mais il y a beaucoup de choses sur les oiseaux qui ressemblent encore à un dinosaur. |
| 27 | 600 | fr | 55.9 | Hershey et Château utilisaient des phages, ou des virus, pour implanter leur propre gène dans un bactérie. |
| 27 | 575 | fr | 88.4 | Mais il y a beaucoup de choses sur les oiseaux qui ressemblent encore à un dinosaur. |
| 28 | 600 | fr | 76.5 | Hershey et Chase ont utilisé des phages, ou des virus, pour implanter leur propre gène dans un bacillus. |
| 28 | 575 | fr | 88.4 | Mais il y a beaucoup de choses sur les oiseaux qui ressemblent encore à un dinosaur. |
| 29 | 600 | fr | 78.8 | Hershey et Chase ont utilisé des phages, ou des virus, pour implanter leur propre gène dans un bactérie. |
| 29 | 575 | fr | 100.0 | Mais il y a beaucoup de choses sur les oiseaux qui ressemblent encore à un dinosaure. |
| 30 | 600 | fr | 78.8 | Hershey et Chase ont utilisé des phages, ou des virus, pour implanter leur propre gène dans un bactérie. |
| 30 | 575 | fr | 88.4 | Mais il y a beaucoup de choses sur les oiseaux qui ressemblent encore à un dinosaur. |
| 31 | 600 | fr | 78.8 | Hershey et Chase ont utilisé des phages, ou des virus, pour implanter leur propre gène dans un bactérie. |
| 31 | 575 | fr | 100.0 | Mais il y a beaucoup de choses sur les oiseaux qui ressemblent encore à un dinosaure. |
| 32 | 600 | fr | 78.8 | Hershey et Chase ont utilisé des phages, ou des virus, pour implanter leur propre gène dans un bactérie. |
| 32 | 575 | fr | 57.2 | Mais il y a beaucoup de choses sur les oiseaux que pourraient encore ressembler à un dinosaures. |

Table 8: 5-prompts without instructions, masking layer by layer of GPTNEO2.7B

## B  LIMITATIONS

This section describes limitations associated with this work.

- Although we have studied three models of two different sizes, 3B and 7B parameters, the 7B parameter model is an instruction-tuned model. Therefore it is not fair to draw any conclusions based on size, but we can draw interesting comparisons between raw and instruction-tuned models.

- We did not identify syntatic or semantic attention heads, or neuron level functionality as is common in interpretability studies. Our research question is on locating rather than analysing individual head's functionality, and decomposing the prompt context.

- We did not conduct multiple rounds of training attention head gates with $L_0$ and results may be slightly different across different random seeds, although we believe the overall conclusion should largely be the same.

- We did not study low-resource language translation, document level translation or other sequence to sequence tasks.

