# OpenReview forum: "Where Does In-context Machine Translation Happen in Large Language Models?"
_ICLR.cc/2024/Conference — Submitted to ICLR 2024_

### Official Review · Reviewer_JM59 · 2023-10-31

**Soundness:** 4 excellent
**Presentation:** 4 excellent
**Contribution:** 3 good
**Rating:** 6
**Confidence:** 4

**Summary:**

While large language models (LLMs) show in-context learning capability, our understanding is still quite limited. This paper explores where the in-context ability occurs in LLMs with machine translation as the testbed. The authors adopt a layer-wise context-masking method on GPTNeo and Bloom, discovering a "task-recognition" point in LLMs beyond which attending to the context is less significant. Further analysis reveals that finetuning layers around such a point are most effective and that layers after this point show higher redundancy.

**Strengths:**

1) The analysis in this paper is intuitive and easy to follow, and is based on two different language models;
2) The findings are very interesting, which show some insights about how the in-context capability evolves in LLMs and may benefit works on inference efficiency and sparsity modeling;

**Weaknesses:**

Experiments are limited to a few language pairs and LLMs of one size, making the generality of the findings questionable.

**Questions:**

While the findings are interesting, I have concerns about how generalizable they are.

Firstly, the authors picked En<->Fr as the translation task, but En<->Fr is often regarded as a relatively easy task due to high language similarity and minimal reorderings. It would be great to have experiments on other languages as well, such as En<->De.

Secondly, the findings are only based on LLMs of one size, i.e. 3B. It's unclear whether readers could expect similar findings on other model sizes.

Besides, based on Figure 1, retaining access to instructions pushes the task-recognition point to earlier layers. Does this also apply to prompt examples? For example, 5-shot prompting often outperforms 1-shot prompting. Can we get an early task recognition point by retaining access to 1 prompt example? If so, we might retain the 5-shot prompting performance but only attend to 1 example for most layers, indicating a high inference efficiency.

Lastly, the statement "the most prunable heads occur after task recognition" might be inadequate. In Figure 5, the head distribution for BLOOM is quite random.


==== After Author Response
Thanks for the response and new experimental results. It seems that the findings could generalize to other language pairs and models to some extent, though the pattern in LLama shows some differences. I tend to keep my positive score.

---

> ### Author Response · Authors · 2023-11-22
> **Author Response to JM59**
>
> We would like to thank the reviewer for their helpful comments and suggestions on our paper.
>
> 1. **Generalisability of Results.** Please see the general response and our updated appendix, where we include additional results on more language pairs and a larger model. We highlight that our conclusions hold across language pairs. For model size we experimented with an Instruction-tuned model, which shows different behavior from the “raw” (not-tuned) ones.
>
> 2. **Distribution of Attention heads.**
> > “In Figure 5, the head distribution for BLOOM is quite random.”:
>
> While it is certainly the case that the masks learned from L0 regularization are much more evenly distributed than those learned from GPTNeo, we emphasize that they are not completely random: roughly 60% of the masked attention heads occur in the last 30% of the model layers (layers 20-30). Furthermore in Figure 10, we note that masking patterns of fr->en and es->en appear quite similar. The similar pattern was reached with different initialisation and different language direction, lending confidence that the masking pattern could be quite consistent.

---

### Official Review · Reviewer_DzYR · 2023-10-31

**Soundness:** 2 fair
**Presentation:** 2 fair
**Contribution:** 2 fair
**Rating:** 3
**Confidence:** 3

**Summary:**

The paper provides insightful findings that GPT models locate, or learn, the translation task at a specific layer during in-context learning. This paper seems extremely interesting to read but the presentation is hard to follow. The set-up is a bit not optimal to me. Meanwhile some finding is not that ``solid". I am happy to consider reading it again and adjust my score if the authors revise the paper to make it easier to follow, plus answering several questions I asked below.

**Strengths:**

* The paper provides insightful findings that GPT models locate, or learn, the translation task at a specific layer during in-context learning. I found the very fascinating and want to learn more about them.

**Weaknesses:**

While I like the work, I don't think it is at the bar of ICLR at the moment. I wanted to point out 3 biggest weaknesses to me - the presentation, the set-up and the conclusion.

* The biggest weakness of the paper to me is that the presentation. I was very excited to read the work but later on I realize it is not that easy to follow the paper at all. I think the difficulty in following the work lies in several parts, to name a few as follows. First, the notations of −, ◦, • as well as Instr−Ex− are somewhat hard to follow and remember to me. I often have to go back and forth to understand this more. Second, there are difficult sentences/paragraphs (examples as below that I could not follow well). Third, I totally get lost at Figure 3 (what are instr_L1L2? what are instr_none?). Fourth, I have an issue with the section of 4.2, while this is nice "Using the example of GPTNeo (32 layers), with a prompt size of 10, and lr = 20, the savings are approximately 35%.", there is no detailed comparison of translation accuracy at all.


* Context masking before a certain level of task- recognition results in a translation model that is often worse than the baseline which sees no instructions or examples. -> I have an issue with this sentence and I always want to take this chance to talk about my concern for the set-up. I consider this is not a surprising fact at all given that the prompt and translation examples appear BEFORE the input. Because of that masking them would obviously distort the representation of the input anyway. It maybe odd but I think the best set-up experiment for this is as follows:
Translate this input: input from english to french, given the following translation examples: ....
I think with this set-up, masking the prompt (from english to french, given the following translation examples: ) and the examples (....) will NOT make that much impact to the translation.

*"Contrary to common fine-tuning wisdom, we show that fine-tuning of model layers is most beneficial in layers associated with task recognition, i.e. the middle layers of the model." -> I found this argument is very weak. The paper already mentions that "Note that this setup is designed to tune the layers for task location. It is highly unlikely that the model can learn translation knowledge from this small amount of supervision", so we cannot take results from such a set-up to say the above finding is correct.


Other notes:
* Interestingly, there is a strong inter-dependency amongst a sequence of layers, where masking out the context results in very poor task recognition. -> I really get lost at this sentence

* Finally, we see that when instructions are masked with the examples (Instr•Ex•), model behavior closely follows that of models which do not receive instructions in their context at all (Instr−Ex•), suggesting that instructions are largely ignored when the examples are present. However, if the model is given complete access to instructions (Instr◦Ex•), it can use the intermediate processing of examples to achieve "task recognition" earlier. -> Somehow this paragraph is also not clear to me.

**Questions:**

* For the analysis in section 4, I understand that the prompt is "Translate English to French." I wonder the case where we change the prompt, e.g. something as follows: "Below are different examples for the translation of English to French. Use them and translate the following input". In the case of different prompt, I was wondering if the specific layer where the plateaus appears (e.g. 20 for GPTNEO, 25 for BLOOM) will change?
* Please elaborate more on Figure 3 about instr_none, intrst_l1l2

---

> ### Author Response · Authors · 2023-11-22
> **Author Response to DzYR**
>
> We would like to thank the reviewer for their detailed and insightful suggestions.
>
> 1. **Notations.** We have clarified the notations to be much more explicit and removed confusing notation, such as replacing `instr_L1L2` with proper terms and removing the usage of $^{\circ,\bullet}$ to be extremely explicit about what is presently masked. Please see general response.
>
>
> 2. **Section 4.2 on Inference Efficiency.**
> > there is no detailed comparison of translation accuracy at all.
>
> The translation accuracy that we should have mentioned, is with respect to the ceiling score that we had observed in the context masking experiments (Fig 1). We have proceeded to clarify in writing "Using the example of GPTNeo ($32$ layers), we see from Fig1 that the model is very close to it's ceiling score after processing the examples at layer 20"
>
> 3. **Discussion on distortion/interdependency of layers.**
> > “masking them would obviously distort the representation of the input anyway. “
>
> We acknowledge that this could be obvious to a reader, and have removed the word “interesting”, and removed the confusing sentence about “interdependency of layers”. Instead we have added more useful discussion around the Llama7b-chat model. We report that the distortion happens strongly when instructions are masked, and there is hardly any distortion when the examples are masked.
>
> 4. **Revise confusing writing.**
> >  The paper already mentions that "Note that this setup is designed to tune the layers for task location. It is highly unlikely that the model can learn translation knowledge from this small amount of supervision", so we cannot take results from such a set-up to say the above finding is correct.
>
> We have revised the sentence to be much more cautious. We revise the claim to be “We show that it is sometimes beneficial to target middle layers of the model which could be associated with task recognition.”
>
> > “Finally, we see that when instructions are masked with the examples...”
>
> We acknowledge that this was difficult to read, we have rewritten this to “Finally, we see that \newtext{the behavior of the model is similar when instructions are masked ($\overline{\texttt{Instr,Ex}}^{Mask}$) and when no instructions are present  ($\overline{\texttt{Ex}}^{Mask}$).} “
>
> 5. **Question about Instructions and Plateu**.
> > In the case of different prompt, I was wondering if the specific layer where the plateaus appears (e.g. 20 for GPTNEO, 25 for BLOOM) will change?
>
> It's an interesting question. We expect that the plateau would not appear at drastically different layers based on what we learnt from the Experiments from Section 4.1, Figure 2. Those experiments showed that even under different number of examples shown, the plateau is reached at almost exactly the same layer. We think that it is unlikely that having a different set of examples would change the plateau point.
>
> Unfortunately we were not able to test this intuition in time for the rebuttal, but would put them in for the next revision (if accepted or not).

---

### Official Review · Reviewer_HfyK · 2023-10-31

**Soundness:** 2 fair
**Presentation:** 3 good
**Contribution:** 2 fair
**Rating:** 6
**Confidence:** 4

**Summary:**

The paper studies in-context learning of MT tasks in LLMs, with a view towards identifying where (at which layers) in-context MT occurs in GPT-style LLMs and characterizing the extent to which layers are redundant for the task in question.  Using GPTNEO2.7Band Bloom 3B for en <-> fr translation on FLORES, the authors explore combinations of instruction prompts and in-context examples and analyze model behavior by layer-from context masking. The authors find that 'task recognition' occurs in middle layers and that masking context earlier significantly disrupts model performance and access to instructions can encourage earlier task recognition. One implication is that increased efficiency can be achieved by masking context (avoiding computational overhead) starting at a certain layer. Additionally, LoRA fine-tuning applied to different layers showed that fine-tuning earlier-to-middle layers is more important than fine-tuning later layers.  The authors then apply the same masking strategy to attention heads, with inconclusive results.

**Strengths:**

The paper succeeds in shedding more light on the phenomenon of in-context learning for MT, specifically the importance of specific layers. Results are intuitively plausible though not entirely surprising. The interpretation of early-to-middle task recognition layers is supported by multiple series of experiments. The methodology used could be applied to other NLP tasks, and implications regarding efficient inference and design of fine-tuning strategies should be of broad interest.

**Weaknesses:**

The paper could have been strengthened by evaluating on more MT tasks, including different language pairs (e.g. high-resource and low-resource pairs) and different complexity (e.g., sentence-level vs. document-level translation).
The section on the importance of attention heads seems fairly preliminary as conclusions differ for the two models investigated. Rather than the model architecture, the training strategy or training data set may play a role here as well (e.g., instruction tuning), there should be a more in-depth discussion of this.
With respect to importance of attention heads for in-context learning, the study by Bansal et al 2022 (arXiv preprint arXiv:2212.09095) may be relevant here.

**Questions:**

1. Do you have any interpretation of the zig-zagging of the green curve in Fig. 1 for  GPTNeo prior to reaching stability?

---

> ### Author Response · Authors · 2023-11-22
> **Author Response to HfyK**
>
> We would like to thank the reviewer for their helpful comments and suggestions on our paper.
>
> 1. **Experiments on additional Language Direction** (en<->es). Regarding the concern of our limited MT tasks, as well as the impact of model training, we would like to direct the reviewer to our general response and the appendix, where we address these concerns with new experimental results. We highlight that the overall trends we observe are prevalent in these new experiments as well. For other tasks such as document level translation, we have proceeded to acknowledge this as a limitation of this study in Appendix Section B.
>
>
> 2. **Discussion on Attention-head masking (Section 6)**
> > “The section on the importance of attention heads seems fairly preliminary as conclusions differ for the two models investigated”:
>
> In Figure 10 (new), we note that masking patterns of fr->en and es->en appear similar. The similar pattern was reached with different initialisation and different language direction, lending confidence that the masking pattern could be quite consistent for translation. Furthermore, while the learned BLOOM masks are much more distributed across model layers than those of GPTNeo, we emphasize that the high-level trend is similar: the majority of masks occur in the last 30% of model layers (layers 20-30) in both models.
>
> 3. **Added Citation.** We thank the reviewer for noting the missed discussion of Bansal et al 2022, which is relevant to our section 6 and which have adapted our discussion to cite.
>
> 4. **Question about Fig1 GPTNeo**
> > Do you have any interpretation of the zig-zagging of the green curve in Fig. 1 for GPTNeo prior to reaching stability?
>
> One possibility that we thought of was that the masked out layers which resulted in the zig-zagging were tightly coupled with the previous layer. This could lead to corruption of the intermediate representations. However, this is mostly speculation and we did not investigate further because it did not occur in the other models, although we know it happens consistently across both translation directions of en $\leftrightarrow$ fr and en $\leftrightarrow$ es for GPTNeo.
>
> 5. **Instruction Tuning**
>
> We have added an additional section discussing Instruction Tuned models (LLama7b-chat) in Appendix A.4 Figure 8. We find that In contrast to GPTNeo and Bloom, the Llama model has no difficulty performing translation when the examples are masked, as long as instructions are present ($\texttt{Instr}\overline{\texttt{Ex}}^{Mask}$). However, when instructions are masked ($\overline{\texttt{Instr}\texttt{Ex}}^{Mask}$), the model performance collapses drastically. This can be explained by the reliance on Instructions for an Instruction-tuned model (\autoref{fig:context_mask_llama7bchat_fren}).}

---

### Official Review · Reviewer_tYJG · 2023-11-01

**Soundness:** 3 good
**Presentation:** 2 fair
**Contribution:** 2 fair
**Rating:** 5
**Confidence:** 4

**Summary:**

The paper is an study of where does the "in-context" Machine translation happens on LLMs.
With that goal in mind,  authors analyze 0-shot and few-shot prompts with and without instructions.
Then they perform several ablation studies to understand when the models generate a task embedding from where there is no degradation.
This is done in several ways:
* masking the context and/or the few-shot examples form one layer onward
* masking the full attention on some layers
* masking attention in all tokens
* studying the sparseness pattern of attention heads via L0 reg ft
* some LORA finetunning.

They try to models GPTNeo and BLoom.

**Strengths:**

They tackle a very difficult and challenging problem and attempt to decipher some conclusions out of the process.
They methodically run experiments with 2 models to attempt to obtain generality.
Their findings are interesting.
The paper is well written and mostly understandable.

**Weaknesses:**

The paper exhibits certain limitations in its experimental approach, which consequently restrict the breadth of its conclusions. The exclusive reliance on the en-fr language pair raises concerns about potential biases. It remains unclear if the observed behavior would persist in en-sp or en-zh pairs or in direct transitions like fr-ge. By analyzing only two LLMs, both with a similar scale of 2.7B and 3B, the conclusions' applicability gets limited as well. This limitation becomes especially pertinent given that other studies, as acknowledged by the authors, like Wei et al. (2023) who suggest that In-Context Learning (ICL) procedure may be model size-dependent.

The paper's predominant focus on the attention mechanism during its ablation study might inadvertently introduce alternative explanations for task embedding, such as the potential for it to merely represent a semantic compression of input tokens. It is not clear which is the alternative research hypotheses to the task vector explanation and the found layer-wise behaviour.
For instance, one could have used a misleading instruction like 'write a summary in French', or 'plan how to solve the enumerated task in English'; or used several alternative instructions to perform the translation task, e.g. "how can I say  '{{sentence}}' in French ?", "what is the English translation of "{{sentence}}" ? , ... etc.

While Figure 1 and Figure 2 seem to display congruent patterns, it's challenging to reconcile them with Figures 3 and 4. Figure 3, in particular, is ambiguous;  the legend terms, like 'instr_L1L2' and 'instr_none', are not clearly defined. Similarly, the patterns oscillate and it is difficult to grasp conclusions w/o another task baseline finetuning.  Figure 4's alignment with the study's conclusions is also complicated due to the presence of layers exhibiting unexpected behavior. the main difference with Figure 1 is that  attention is limited to all other tokens, including the partial decoding and the input to translate. Maybe another interesting experiment is to mask attention from all input (not only instructions and examples), this could help understand this picture better or at least it will help to get a better hypotheses of why masking some initial layers. It might be that the task layers are simply compressing the input similarly to an encoder-decoder architecture and then the upper layers are simply decoding for which incrementally refining the hypotheses can provide limited gains.

Regarding  the L0 regularization optimization, there might be several masks that can lead to similar solutions with different patterns. It would have been nice to see what happens if one biases the L0 masking loss towards both the "task identification layers" and the non-relevant layers. This modification would have been a stronger evidence towards the conclusion.

Additionally, conducting analogous experiments in other domains, such as summarization, would undoubtedly offer a more comprehensive perspective specially for figure 1 and figure 2.

Lastly, the use of "Instr-Ex*" nomenclature was frequently perplexing, especially when it often seems synonymous or expandable in the same manner as 'instr_L1L2'.

**Questions:**

Following the week points of the paper:
* do the findings generalize to other languages ?
* should we expect similar behaviour on larger LLMs ?
* can the transition of layer mechanism be understood as "encoder"-like compression and "decoder"-like refinement ?
* Would different instructions to achieve translation generate different pictures ?
* how relevant are the specific examples selected in few-shot example ?
* would other tasks behave similarly ?
* where are inputs stop being used ? What is the missing plot between Fig 1 and Fig 4 -- if that included all layers above a given layer --
* are the L0 masks unique ? can we bias them against our conclusion and still mask 30 % of the heads ?

---

> ### Author Response · Authors · 2023-11-22
> **Author Response to tYJG**
>
> We would like to thank the reviewer for their thoughtful comments and questions.
>
> 1. **Language Pairs and Model Sizes.** We direct the reviewer to the general response, where we address these concerns, and to the updated appendix where we include our results from these new experiments. We highlight that our conclusions hold across language pairs. For model size we experimented with an Instruction-tuned model, which shows different behavior from the “raw” (not-tuned) ones.
>
> 2. **Notation.** Similarly, we have updated our draft to include changes to the notation mentioned, such as replacing `instr_L1L2` with proper terms and removing the usage of $^{\circ,\bullet}$.
>
> We now address concerns that are specific to reviewer tYJG.
>
> 3. **Experiment masking attention from all input.**
>
> > “Maybe another interesting experiment is to mask attention from all input (not only instructions and examples)”:
> > where are inputs stop being used ? What is the missing plot between Fig 1 and Fig 4 -- if that included all layers above a given layer --
>
> We do run this proposed experiment in Appendix A2 of the original draft (A5 /Figure 9 in the updated draft), on the en$\rightarrow$ fr language pair for both GPTNeo and BLOOM. We found that a model’s ability to perform the task diminishes _much faster_, layerwise, when we mask the entire input, rather than just the context, suggesting that attention to the input is still necessary after we have masked out the context. We have added a discussion of these results in the appropriate appendix section (A5).
>
> 4. **Discussion reconciling all experiments.**
> > “While Figure 1 and Figure 2 seem to display congruent patterns, it's challenging to reconcile them with Figures 3 and 4”:.
>
> On a high level, we can say quite confidently that there appears to be task recognition or critical task layers happening at around the same middle layers of the model. Fig1 and 2 shows task recognition after a certain layer, Fig 3 shows masking of layers across the model is not equal, and Fig4 shows attention heads in later layers can be masked. However while we have tried to explore the title question from multiple angles, the specific details could indeed be hard to reconcile and we hope this work can inspire a deeper investigation.
>
> 5. **L0 Regularization Optimization.**
> > “Regarding the L0 regularization optimization, there might be several masks that can lead to similar solutions with different patterns.”
>
> In Figure 10 (new), we note that masking patterns of fr->en and es->en appear quite similar. The similar pattern was reached with different initialisation and different language direction, lending confidence that the masking pattern could be quite consistent.
>
> Also, while it is true that other masking schemes may yield different solutions, we note that L0 masks are not biased towards task identification layers. Instead, the masking scheme we leverage acts as a method to determine which attention heads are most important to in-context MT, and which can be discarded.
>
> 6. **Question about few-shot examples**
> > how relevant are the specific examples selected in few-shot example ?
>
> In our main paper experiments, for few-shot examples we have repeated the experiments with 5 random seeds to remove the effect of specific examples.
>
>
> 7. **Question about Different Instructions**
> > Would different instructions to achieve translation generate different pictures ?
>
> We expect that the plateau would not appear at drastically different layers based on what we learnt from the Experiments from Section 4.1, Figure 2. Those experiments showed that even under different number of examples shown, the plateau is reached at almost exactly the same layer. We think that it is unlikely that having a different set of examples would change the plateau point.
>
> Unfortunately we were not able to test this intuition in time for the rebuttal, but would put them in for the next revision (if accepted or not).
>
> 8. **Question about Other Tasks**
> > would other tasks behave similarly ?
>
> We are not sure how other tasks should behave, and have 1) listed this as a limitation of this study 2) are careful not to overclaim the contributions of this study.

---

### Author Response · Authors · 2023-11-22
**General Response to All Reviewers**

We’d like to thank all reviewers for their invaluable feedback on our work. We have worked to revise the paper according to the feedback. Below, we highlight the major revisions:

1. **Experiments on additional Language Direction (en<->es); Appendix Figure 7.** Several reviewers (tYJG, JM59, HfyK) raised concerns with our experiments being limited to only English to French and French to English translation. To address this concern, we have run our core experiments on English to Spanish and Spanish to English language pairs as well. Overall, we see nearly identical trends to what we observe in the English and French experiments, suggesting that our findings generalize across other languages.

2. **Experiments on different model capacities (Llama7b-chat), Appendix Figure 8.** Another concern raised by multiple reviewers (tYJG, JM59, HfyK) is that our experiments are limited to two with distinct pre-training but similar capacity. To address this concern, we also include experiments on the en -> fr language direction using llama-7b-chat, which is both a larger model and instruction-tuned after pre-training with both supervised datasets and RLHF in the Appendix (Figure 8).

3. **Clarification in notations.; Table 1, Figure 3, and Appendix Figure 6** Finally, another concern raised by all reviewers was the lack of clarity around the notations used for some of our experimental setups. In Figure 3, we altered the legends to explicitly declare [With Instructions, No Instructions], rather than the previous notation which was not clear. Additionally, we have changed the notation used for our section 4 experiments; rather than using $^{\circ, \bullet, -}$, we we use an overline notation, e.g. $\overline{\texttt{Ex}}^{Mask}$, to make the format of the context more clear and added Figure 6 in the appendix for additional visual description.


All of our edits to the draft are made in blue.

---

### Meta-Review · Area_Chair_AfCx · 2023-12-05

**Metareview:**

The paper studies when “in-context” machine translation happen in LM, by offering a layer-wise analysis. Specifically, they look into “task-recognization” phase and input-output mapping phase. The paper is mostly well-written, and experiments are solid (done in two language models), and findings are interesting!). I have read the review contents rather than simply looking at the scores, disregarding reviewer DzYR's concerns with presentation. However, I still think the paper will benefit from another round with a bit more thorough evaluation of applying analysis to models with different scale (reviewer tYJG's comment). The authors have added results on instruction-tuned LLAMA results during the rebuttal which shows different trend compared to other models they have studied. I think the paper will benefit from studying this a bit further, e.g., comparing with non-instruction tuned LLaMA model as well, and expand on the discussion of how scale impacts the analysis.

**Justification For Why Not Higher Score:**

The experimental scope is somewhat limited, only considering high resource language pair. More experiments on considering different scale model will be important given ICL is a problem where scale leads to substantially different model behavior (reviewer tYJG’s comment).

**Justification For Why Not Lower Score:**

The research question is interesting and experiments are carefully done, with interesting findings.

---

### Decision · Program_Chairs · 2024-01-16

Reject